# Mechanism of ATP hydrolysis in the Hsp70 BiP nucleotide-binding domain

Guillaume Mas ⓘ & Sebastian Hiller ⓘ ✉

The 70 kDa heat shock protein (Hsp70) family of molecular chaperones ensures protein biogenesis and homeostasis, driven by ATP hydrolysis. Here, we introduce *in-cyclo NMR*, an experimental setup that combines high-resolution NMR spectroscopy with an ATP recovery and a phosphate removal system. In-cyclo NMR simultaneously resolves kinetic rates and structural information along functional cycles of ATP-driven molecular machines. We benchmark the method on the nucleotide binding domain (NBD) of the human Hsp70 chaperone BiP. The protein cycles through ATP binding, hydrolysis, and two parallel pathways of product release. We determine the kinetic rates of all eleven underlying elementary reactions and show these to match independent measurements. The two product release pathways regulate the cycle duration dependent on the products concentration. Under physiological conditions, they are both used. The in-cyclo NMR method will serve as a platform for studies of ATP-driven functional cycles at a remarkable level of detail.

The 70 kDa heat shock protein (Hsp70) family of molecular chaperones is crucial for biogenesis and protein homeostasis[1–4]. Hsp70s account for up to 4% of total cellular protein mass, making them one of the most abundant proteins in the cell[5–7]. They are involved in diverse cellular processes[3,8–12], including de-novo protein folding at the ribosome[13,14], protein translocation through pores[15,16] and solubilization of protein aggregates[1,17,18]. Consistently, Hsp70s are connected to multiple patho-physiological conditions including cancer and neurodegenerative diseases[19–21]. In the endoplasmic reticulum (ER), BiP (immunoglobulin Binding Protein) is the sole Hsp70 isoform[22,23] and the most abundant ER chaperone[7,24]. BiP is the central functional hub of the ER chaperone network that ensures protein folding homeostasis in the "folding factory of the cell". It consequently binds to most of the proteins that are processed in the ER, promoting their folding and preventing their aggregation[25,26]. Additionally, it also acts as the central regulator of the unfolded protein response (UPR) by binding to the UPR sensors in a stress-dependent manner[27–29]. Furthermore, BiP targets unfolded proteins to the degradation machinery associated with the ER-associated degradation pathway (ERAD)[30–32]. Moreover, BiP is overexpressed in many human cancers, making it a major therapeutic target[33–36].

Hsp70 chaperones consist of two distinct domains, the nucleotide-binding domain (NBD) and the substrate-binding domain (SBD), which are connected by a flexible linker[37,38]. The NBD has a clamp-like shape with two lobes I and II[12]. Each lobe is subdivided into two subdomains A and B. A nucleotide-binding site is located in the cleft between lobes I and II. Hsp70 chaperones go through a functional cycle encompassing ATP-binding, ATP-hydrolysis, and ADP·Pi release, all of which take place in the NBD[39]. ATP binding leads to a rearrangement of lobe I, which triggers the SBD to open the client binding site[40–42]. Following ATP hydrolysis, the NBD is in an ADP-bound state. From there, ADP is released at one point, resulting in the apo form, to which a new ATP molecule binds to restart the cycle. The overall Hsp70 chaperone activity resulting from this fundamental cycle depends on the cellular context and manifests into a diverse set of effective functions such as a foldase, holdase, translocase, unfoldase or disaggregase[11,12,43]. Importantly, these functions are fundamentally regulated by the timing originating from nucleotide processing in the NBD and understanding the Hsp70 functional cycle requires elucidating the nucleotide reaction steps and their modulation by environmental changes.

Despite the exhaustive biochemical characterization of Hsp70 proteins, measurements of the functional cycle kinetic parameters have so far been possible only by isolating individual steps and in the absence of direct structural observation[44–46]. Recently, NMR spectroscopy has emerged as a powerful method to investigate the molecular mechanism of various ATPases[47–49]. In this study, we develop in-cyclo

Biozentrum, University of Basel, Basel, Switzerland. ✉e-mail: sebastian.hiller@unibas.ch

NMR, a method that combines the power of methyl NMR spectroscopy to resolve atomic sites with a temporal dimension resolving the kinetic parameters of the functional cycle. We benchmark the method on the example of the NBD of the human Hsp70 chaperone BiP from the endoplasmic reticulum and compare the results from the in-cyclo experiments with classical single turn-over and MABA-ADP release experiments. Our analysis reveals that ADP·Pi is released via two parallel mechanisms, either ADP or Pi leaving first. We determine the effect of calcium, a key ER stress marker that regulates the activity of multiple ER proteins[50], on the cycle and assess the role of fluctuating ADP concentration. In-cyclo NMR provides a technology platform for a fundamental understanding of the Hsp70 functional cycle at the atomic level, also in the context of the full-length Hsp70s and their regulation by co-chaperones.

## Results

### NMR resonance assignment of NBD methyl groups

As a first step towards observing individual atomic sites during the functional cycle of BiP NBD, we established a highly pure and homogenous sample preparation. Since we wanted to be able to detect conformational sub-states of the protein with potentially low populations, the preparation needed to be free of bound nucleotides and other contaminants, which are a major cause of concern[51]. We purified the protein using established protocols and then added an affinity column purification step under denaturing conditions of 8 M urea. Under these conditions, the protein is entirely unfolded and thus loses the affinity for bound impurities, which are washed through the column. After elution from the column, the protein was slowly refolded via dialysis. The percentage of purity was determined by SDS-page gel to be > 98% (Supplementary Fig. 1a). The refolded protein was analyzed by SEC-MALS, NMR spectroscopy and an NADH-coupled ATP assay, and compared with protein purified without the unfolding/refolding step (Supplementary Fig. 1b–d). The two NMR spectra overlapped perfectly, demonstrating that BiP NBD reaches its native state after denaturation and refolding. This high purification standard was kept in all subsequent experiments.

In a next step, we isotope-labelled the methyl groups of three amino acids, Ile-[$^{13}C^1H$]$^{\delta1}$, Met-[$^{13}C^1H$]$^{\epsilon}$ and Val-[$^{13}C^1H$]$^{\gamma1/\gamma2}$, on an otherwise deuterated background. This is a well-established technique to allow atomic resolution NMR studies even at large molecular sizes up to several 100 kDa[52]. 2D [$^{13}C,^1H$]-methyl-TROSY spectra[53] with high sensitivity can be recorded in times as short as 5 min at protein concentrations of 100 μM. This high sensitivity is key to detecting also minor sub-states of the cycle when longer experiment times are used. After these purification and preparation steps, NMR spectra of the protein were recorded in the apo form or equilibrated with an excess of a ligand of interest, such as ADP and Pi.

In the presence of 5 mM ADP·Pi, we observed a homogeneous spectrum with a single set of 102 resonances, precisely matching the 102 resonances expected from the chemical structure of the molecule (Fig. 1a). We established sequence-specific assignments of these resonances using a strategy that combines single-point mutagenesis and NOESY experiments. As a first step, we established the assignments of all 6 methionine residues by single-point mutagenesis (M148L, M153L, M196L, M263L, M332L, M339L). Then, using these anchor points, we expanded the assignment using 3D $^{13}C$, $^{13}C$-resolved [$^1H$, $^1H$]-NOESY experiments that we manually curated against a published crystal structure of the BiP NBD (PDB 5EVZ) (Fig. 1b and Supplementary Fig. 2a). The methyl groups of Met, Val and Ile have unambiguously separated chemical shift ranges for their NMR signals, permitting a direct identification of the amino acid type for a given signal and thus to easily distinguish different NOESY networks. We resolved around 4 NOE contacts per residue and restricted ourselves to only those assignments that could be unambiguously made by the network effect. We additionally exploited the good correlation between the NOE cross

peaks intensities and the calculated distance in the BiP NBD structure to confirm the correctness of these networks (Supplementary Fig. 2b). Overall, we could resolve 100 NOESY contacts up to 5 Å, 255 NOESY contacts in the distance range 5–8 Å and 91 NOESY contacts in the range 8–10 Å, leading to a high-confidence assignment (Fig. 1c). As a final validation step, we selected 7 individual residues at the core of large NOESY networks and confirmed the correctness of their assignment by single-point mutagenesis. In total, this approach led to the stereospecific assignment of 60 residues, thereof 32/35 valines, 22/25 isoleucines and 6/6 methionines. Among the assigned 32 valines, 24 had both methyl Cγ1 and Cγ2 assigned and 8 had one of them, resulting in a total of 84 observable NMR signals (Fig. 1a, d, Supplementary Fig. 3). For the subsequent experiments, we thus have 84 atomic-level reporters that we can observe simultaneously and with high sensitivity for local structural changes. We next acquired the NMR fingerprint spectra of four states, apo NBD, Pi-bound NBD, ADP·Pi-bound NBD and ADP-bound NBD, as references to later identify the states populated during the functional cycle (Fig. 2a, b). For each state, we recorded a [$^{13}C,^1H$]-methyl-TROSY spectrum and used titration experiments to transfer assignments from the ADP·Pi-bound state, wherever this was unambiguously possible (Supplementary Fig. 4a–c). We could then also identify the residues that have a different signal in all four states, to serve as reporter residues in future experiments. Examples are residues V50cγ1, I52cδ1 and V172cγ2 (Fig. 2b). Using these NMR titration experiments together with isothermal titration calorimetry (ITC) experiments, we determined the four dissociation constants for ADP, Pi, and ADP·Pi binding, $K_D^2 = 1.6 \pm 0.2\ \mu M$, $K_D^3 = 310 \pm 40\ \mu M$, $K_D^4 = 280 \pm 50\ \mu M$ and $K_D^5 = 0.9 \pm 0.1\ \mu M$ (Supplementary Fig. 4d-g).

### Setup of the functional cycle with the ATP regeneration system

Based on the prerequisites of a highly pure preparation and near-complete resonance assignments in four reference states, we set up the experiment to monitor the BiP NBD functional cycle. We selected a buffer composition corresponding to optimal Hsp70 activity (25 mM HEPES pH 7.5, 150 mM KCl and 10 mM MgCl$_2$)[45,54–56], and the physiological temperature of 37 °C. Magnesium is required for the Hsp70 ATP hydrolysis as it coordinates the ATP β and γ phosphate[57]. The potassium concentration was chosen to match previous reports that it acts as a cofactor of the Hsp70 hydrolysis of ATP increasing the activity by 5-fold in the optimal range of concentration between 100 and 150 mM[58].

Notably, while it is possible to prepare pure apo and pure ADP·Pi-bound states, it is not possible to prepare the ATP-bound state in thermodynamic equilibrium, due to the catalytic activity of the protein. For example, the addition of 5 mM ATP to 100 μM of BiP NBD leads to the rapid accumulation of ADP and free phosphate and results in a non-equilibrium situation with continuously changing ADP:ATP concentration ratio and a continuously increasing phosphate concentration.

In order to create a stable steady-state condition, we implemented two enzymatic systems inside the NMR tube to convert ADP into ATP and to remove the free phosphate, respectively. The first enzyme (E1) is pyruvate kinase, at catalytic amounts, which combines phosphoenol pyruvate (PEP) and ADP to form ATP (Fig. 3a). Monitoring of ATP, ADP, PEP and pyruvate concentrations by 1D $^1H$ NMR spectra shows that the ATP regeneration system keeps the concentrations of ATP and ADP effectively constant over long periods of time with no detectable signal for the ADP (e.g. [ATP] = 5 mM and [ADP] <25 μM) (Fig. 3b, c). Thereby, the PEP concentration is linearly decreasing and the pyruvate concentration linearly increasing with the same rate (Fig. 3d, e). The linear increase of pyruvate corresponds stoichiometrically to the ATP consumption and thus directly allows to determine the ATP hydrolysis rate of the BiP NBD (Fig. 3d,e). The second enzyme (E2) is sucrose phosphorylase, at catalytic amounts, which catalyzes the conversion of sucrose and phosphate into α-D-glucose-1-phosphate and D-fructose. This enzyme thus removes any free phosphate emerging from ATP

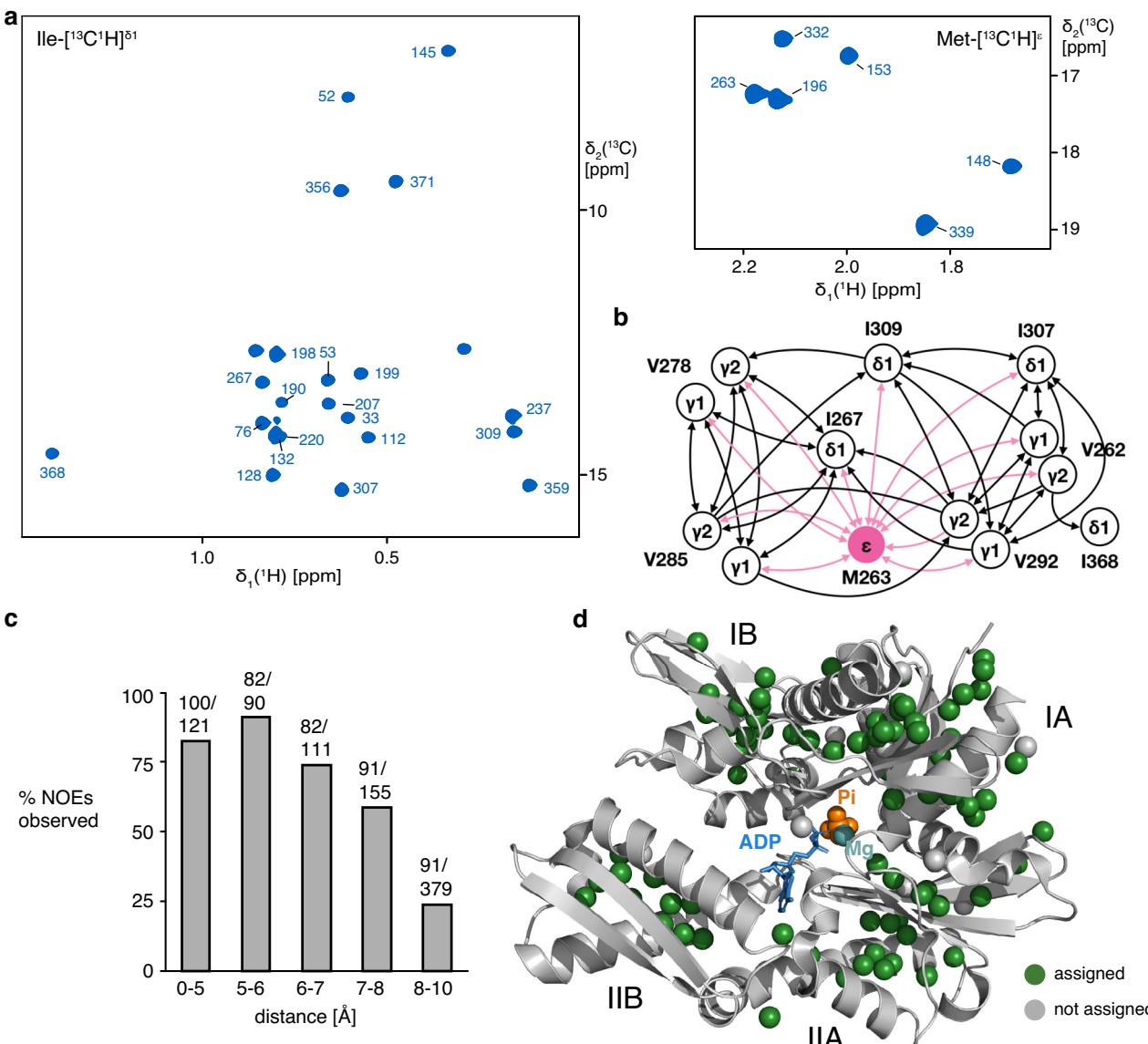

**Fig. 1 | Assignment of the BiP NBD ADP·Pi-bound state. a** 2D [$^{13}$C,$^1$H]-methyl-TROSY spectra of methyl-labeled BiP in the presence of 5 mM ADP·Pi. Sequence-specific resonance assignments are indicated. **b** Graphical representation of the NOE network around methionine 263 (pink). Each arrow tip corresponds to one observed NOE cross peak between two methyl groups. **c** Completeness of the BiP Ile, Met and Val methyl NOE network, as expected from the structure PDB 5EVZ. For each distance range of the interproton distance, the numbers of observed/expected NOEs are indicated, and the percentage given as a grey bar. **d** Location of the assigned methyl groups (green spheres) in a 3D structure (PDB 5EVZ).

hydrolysis by BiP ([Pi] <25 µM)). Monitoring of the glucose-1-phosphate concentration by 1D $^1$H NMR spectra shows a linear increase matching the pyruvate concentration (Fig. 3d, e). To distinguish this experiment from equilibrium experiments, and highlight the unique setup to control ATP, ADP and Pi levels simultaneously, we refer to this setup as the "in-cyclo" NMR experiment. The ATP consumption of 100 µM BiP NBD in the in-cyclo experiment was 0.15 ± 0.01 mM min$^{-1}$, which corresponds to a macroscopic hydrolysis rate of $k_{hydr}$ = 1.47 ± 0.05 min$^{-1}$ (Fig. 3f). The inverse of this rate, $T$ = $k_{hydr}^{-1}$ = 41 ± 2 s is the average length of the functional cycle of BiP NBD. Our assay thus establishes kinetic properties of the cycle, while simultaneously allowing atomic resolution observations.

### Direct observation of the ATP-bound state
In the next step, we employed the in-cyclo setup properties to get atomic-level insights. 2D [$^{13}$C,$^1$H]-methyl-TROSY spectra recorded in-cyclo with active ATP regeneration and phosphate removal showed

that most residues show two distinct NMR signals, while a smaller number of residues show three distinct signals (Fig. 4a). The protein thus populates three detectable states on its functional cycle. To identify the nature of these states, we compared the in-cyclo NMR spectra with the reference spectra. Two of the three states matched perfectly with a reference, thus identifying them as the ADP and ADP·Pi-bound states (Fig. 4a, b and Supplementary Fig. 5). For some residues, the signals of the ADP and ADP·Pi-bound states are identical, and these residues thus feature a single signal in-cyclo that corresponds to the sum of the two individual signals. The third detected state did match neither apo NBD, Pi-bound NBD, ADP·Pi-bound NBD nor ADP-bound NBD (Fig. 4a, b and Supplementary Fig. 5). It was detected only in the presence of the in-cyclo setup or in the early phase of non-equilibrium experiments with pure ATP added. This set of NMR signals therefore corresponds to the ATP-bound state. We term ADP-bound NBD "D state", ADP·Pi-bound NBD "D·Pi state", and ATP-bound NBD "T state". The apo state and the Pi-

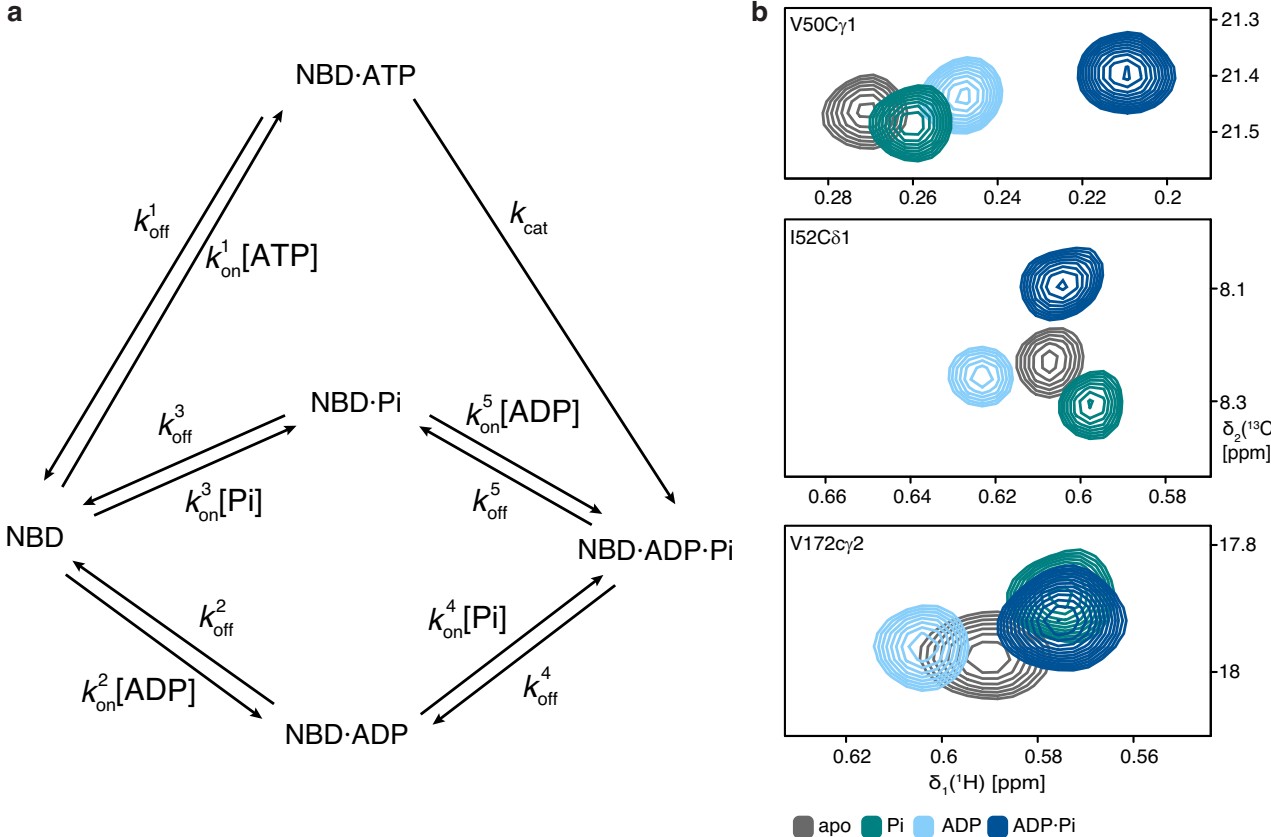

**Fig. 2 | Reaction scheme and NMR fingerprints for the BiP NBD functional cycle.** **a** Scheme of the BiP NBD functional cycle including its kinetic parameters. See theory section for details. **b** Sections of 2D [$^{13}$C,$^1$H]-methyl-TROSY spectra of residues Val50cγ1, I52cδ1 and V172cγ2 for the equilibrium experiments of NBD apo (grey), NBD with 5 mM Pi (green), NBD with 5 mM ADP (light-blue) and NBD with 5 mM ADP·Pi (dark-blue).

bound state were not significantly populated during the functional cycle.

Sequence-specific resonance assignments for the ATP-bound state were established by direct transfer from the ADP·Pi-bound state for all signals in unambiguous spectral regions (Fig. 4a, b and Supplementary Fig. 6) and further validated by the 13 single-point mutants that had been used for the assignment of the ADP·Pi-bound state. In particular, the assignment transfer in spectrally crowded regions was facilitated by stereospecifically labeled valine γ2 to reduce the number of peaks by 50% (Supplementary Fig. 6). In total, 74 unambiguous assignments for the ATP-bound state could be established (Fig. 4a and Supplementary Fig. 6). With the assignments at hand, we analyzed the chemical shift differences between the ATP-bound and the ADP·Pi-bound states. The differences were largest for residues located in the vicinity of the bound nucleotide (Fig. 4c and Supplementary Fig. 7a). Additional large chemical shift differences were however also observed in the lobe IA. These reflect the rotation of the lobe IA resulting from ATP binding, which in the full-length protein results in docking of the SBD to the ATP-bound state[12].

Having established the signatures of different nucleotide states, we were also curious about the effects of the slow-hydrolysable ATP analogs AMPPNP (adenylyl imidodiphosphate), AMPPCP (adenylyl methylenediphosphate) and ATP-γ-S (adenosine 5′-(gamma-thio-triphosphate)). These analogs are frequently used to mimic the ATP-bound state[59], however crystal structures of the NBD in complex with these analogs, in apo form, or in complex with various nucleotides show no significant differences (Supplementary Fig. 7b, c). The assessment of their 2D [$^{13}$C,$^1$H]-methyl-TROSY spectra fingerprints showed that the NMR fingerprints of all three ATP analogs resemble the ADP·Pi-bound state rather than the ATP-bound state

(Supplementary Fig. 7d), i.e., they shift the BiP NBD into a conformation that is more similar to the ADP·Pi-bound state than to the ATP-bound state. For ATP-γ-S, the fingerprint spectrum showed a peak splitting, indicating the formation of a heterogenous mix of conformations (Supplementary Fig. 7d). These observations explain why neither of these three analogs induces the expected Hsp70 interdomain conformational change that is triggered by ATP binding as it has been reported in the literature[44,60–64].

## Combined measurement of the functional cycle kinetic parameters

We then set out to determine the kinetic parameters of the functional cycle. The NMR signal intensities of each of the three detected states are proportional to their population in the steady state. We integrated 51 non-overlapping independent methyl groups to obtain the population ratio $p_{Dtotal}/p_T$, including 7 non-overlapping independent methyl groups to obtain the subpopulation ratio $p_{D·Pi}/p_D$. Distributed across the entire protein, these ratios showed little variation that are mainly due to differences in local dynamics of individual residues, as well as experimental noise. Our usage of average populations, instead of only a few residues, cancels out effects of individual local dynamics resulting in increased precision (Fig. 4d). The relative populations are T: 36 ± 5%, D·Pi: 44.6 ± 4% and D: 19.4 ± 2% (Fig. 4e). Because the average length of a complete cycle is $T = 41 ± 2$ s, these population levels correspond to average occupancy times of $t_T = 15 ± 3$ s, $t_{D·Pi} = 18 ± 2$ s and $t_D = 8 ± 2$ s (Fig. 4f).

In general, the functional cycle goes through a kinetic network of mostly reversible reactions, with a total of 11 free parameters (Fig. 2a, see theory section). Our in-cyclo experiment simplifies the cycle by setting the concentrations of ADP and phosphate in the bulk to below

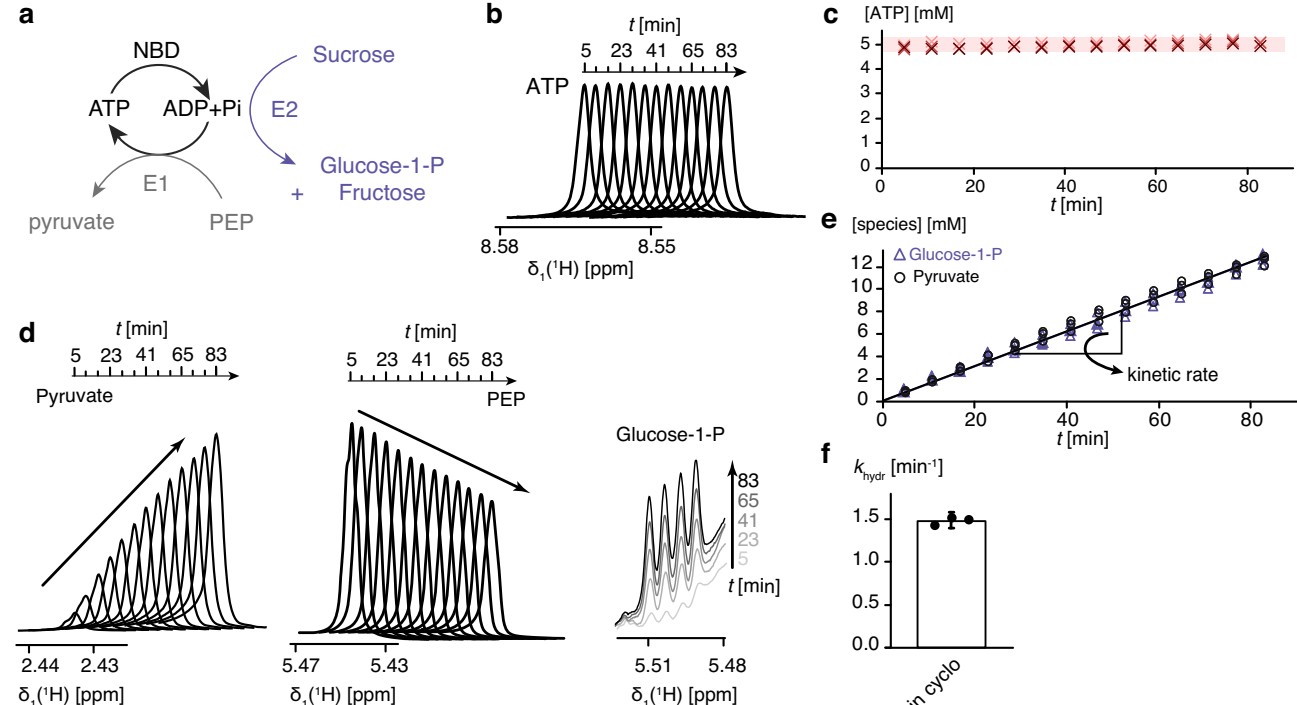

**Fig. 3 | Experimental setup for in-cyclo NMR with ATP regeneration and phosphate removal system. a** Reaction scheme of the ATP regeneration system (E1; pyruvate kinase) and phosphate removal system (E2; sucrose phosphorylase). **b** Series of 1D $^1$H NMR spectra of the ATP indole H8, recorded at an interval of 6 min. **c** Quantification of the ATP concentration from the 1D $^1$H NMR spectra shown in (b): Red crosses represent three independent experiments. **d** Same as (**b**) for pyruvate (left), PEP (middle) and glucose-1-phosphate (right). **e** Quantification of the

pyruvate (black circle) and glucose-1-phosphate (lavender triangle) concentration from the 1D $^1$H NMR spectra shown in (**d**): the symbols represent three independent experiments. The black line shows a linear fit to determine the ATP consumption rate. **f** Molecular hydrolysis rate calculated from the in-cyclo data presented in (**e**). Data points represent three independent experiments. The bar represents mean and standard deviation.

25 µM and thus rendering the back-reactions involving ADP or phosphate binding essentially negligible. The ATP hydrolysis step is irreversible, and the average occupancy time of the T-state thus provides a reliable determination to the catalytic ATP hydrolysis rate $k_{cat} = t_T^{-1} = 0.067$ s$^{-1}$. The average occupancy time of the D·Pi state is equivalent to the sum of two off-rates as $1/t_{D·Pi} = k_{off}^4 + k_{off}^5 = 0.055$ s$^{-1}$.

To fully resolve all kinetic parameters, we employed a data fitting procedure to four experimental data sets, (i) the in-cyclo experiment with ATP regeneration and phosphate removal, (ii) ATP regeneration only, (iii) single batch experiments from pure ATP with the phosphate removal system, (iv) single batch experiments from pure ATP (Figs. 3, 4 (i) and Supplementary Fig. 8a–c (ii, iii, iv). Thereby, the values obtained above were directly used as initial values for the fitting procedure. Because the NBD·Pi-bound state is not significantly populated in-cyclo, we independently determined the kinetic rates of phosphate binding and unbinding, $k_{off}^3 = 1.32 \pm 0.31$ s$^{-1}$ and $k_{on}^3 = 5 \pm 2$ mM$^{-1}$ s$^{-1}$ using an EXSY NMR experiment (Supplementary Fig. 9). With the four experimental data sets, the fitting procedure determined a consistent set of kinetic parameters for all steps of the functional cycle that simultaneously explained all data (Table 1, Supplementary Fig. 10a–d). The solution was numerically stable (Supplementary Fig. 10e). Attempts to determine the kinetic parameters without the data from the in-cyclo experiment did not yield a converging solution.

The data show that the average occupancy of the T state is indeed dominated by the ATP hydrolysis process with $k_{cat} = 0.065 \pm 0.013$ s$^{-1}$, whereas ATP unbinding is a factor of 5 times rarer with $k_{off}^1 = 0.012 \pm 0.003$ s$^{-1}$. ATP hydrolysis directly results in the ADP·Pi-bound state. The dissociation of this state can occur via two possible pathways, either ADP or phosphate unbinding first. Thereby, the rate of phosphate release is 2.5 times faster than ADP release. Then, release

of phosphate from the Pi-bound state is so fast that this state is invisible in the in-cyclo experiment. In contrast, release of ADP from the NBD·ADP complex is sufficiently slow such that this state is detected. Either of the two dissociation pathways results in the apo form, from which ATP binding starts a new cycle.

## Comparison of kinetic parameters with independent measurements

We next set out to benchmark these kinetic parameters to independently measured values. Out of the 11 free parameters of the functional cycle, 3 parameters can be commonly accessed with classical experiments ($k_{cat}$, $k_{off}^2$ and $k_{off}^5$). We benchmarked $k_{cat}$ with standard single-turn over experiments[44,46]. For these experiments, the NBD·ATP complex is initially formed by incubation of an excess of ATP with BiP NBD at 4 °C. At this temperature, it is generally assumed that the hydrolysis rate can be neglected. Unbound nucleotides were removed by gel filtration and the purified NBD·ATP complex was flash frozen. The complex was then incubated at 37 °C for variable time, after which the reaction was quenched by addition of HCl. This step unfolds the NBD, thus immediately stopping the reaction and releasing all educts and products into solution. From the resulting samples, the [ATP]/[ADP] ratio was determined by anion exchange chromatography[65]. The data were fitted to a monoexponential, resolving the ATP half-life time that corresponds to a $k_{cat}^* = 0.071 \pm 0.02$ s$^{-1}$ (Supplementary Fig. 11a, b). Because $k_{cat}^*$ can be similar or slightly larger than $k_{cat}$ (see theory section), these single turn-over experiments showed an excellent agreement with the fitted $k_{cat} = 0.065 \pm 0.01$ s$^{-1}$ from the in-cyclo experiment.

We next wanted to compare our determined kinetic rates of ADP dissociation, $k_{off}^2$ and $k_{off}^5$, with a standard assay in the Hsp70 field that

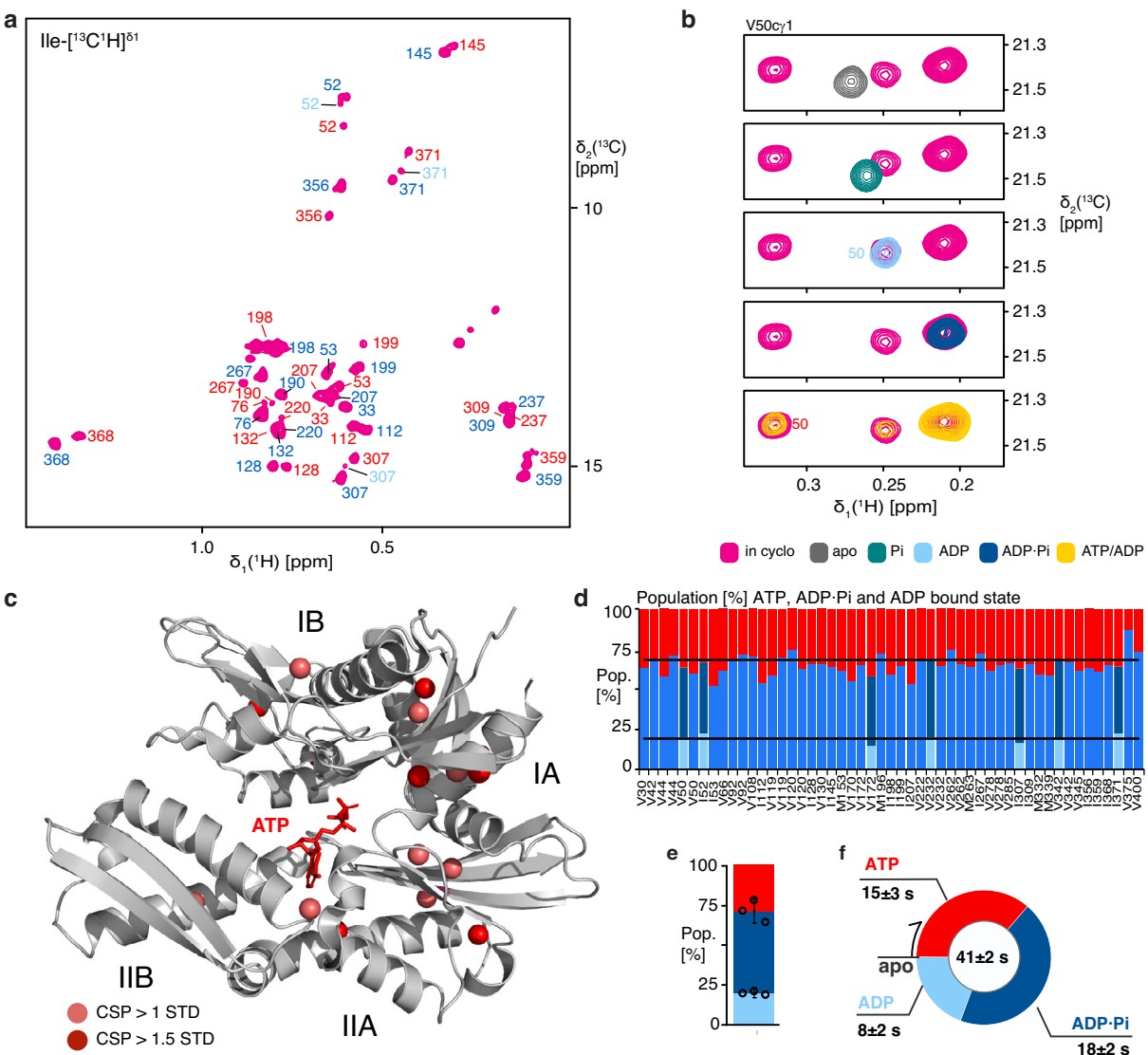

**Fig. 4 | Resonance assignment of the NBD·ATP-bound state. a** Sections of 2D [$^{13}$C,$^1$H]-methyl-TROSY spectrum of methyl-labeled NBD in presence of the ATP regeneration system (in-cyclo). Sequence-specific resonance assignments of the Ile-[$^{13}$C$^1$H]$^{δ1}$ methyl groups in the ADP-bound (light-blue), ADP·Pi-bound (dark-blue) and ATP-bound (red) states are indicated. **b** Selected sections of 2D [$^{13}$C,$^1$H]-methyl-TROSY spectra of residue Val50cγ1 in-cyclo (pink) compared to the equilibrium experiments of NBD apo (grey), NBD with 5 mM Pi (green), NBD with 5 mM ADP (light-blue), NBD with 5 mM ADP·Pi (dark-blue) and NBD with 5 mM ATP (yellow). **c** Methyl groups with significant chemical shift differences between the ATP-bound

and ADP·Pi-bound states displayed on the structure of the ATP-bound state (PDB 3LDL). The CSPs are shown in Supplementary Fig. 7a. **d** Residue-specific populations of the ADP-bound (light-blue), the ADP·Pi-bound (dark-blue) and the ATP-bound state (red). Each bar corresponds to one methyl group. (black line = average populations). **e** Populations of the ADP-bound, ADP·Pi-bound and ATP-bound states, averaged over all resonances. Data points represent three independent experiments. The bar represents mean and standard deviation. **f** Average length of the functional cycle of the NBD functional cycle including average state occupancy time.

**Table 1 | Kinetic parameters of the BiP NBD functional cycle at standard buffer conditions and T = 37 °C**

| label | reaction | $K_D$ μM | $k_{on}$ [mM$^{-1}$ s$^{-1}$] | $k_{off}$ [s$^{-1}$] | $k_{cat}$ [s$^{-1}$] |
|---|---|---|---|---|---|
| 1 | NBD·ATP ⇌ NBD + ATP | 0.38 ± 0.12 | 32 ± 9 | 0.012 ± 0.003 | |
| 2 | NBD·ADP ⇌ NBD + ADP | 1.43 ± 0.30 | 77 ± 28 | 0.11 ± 0.04 | |
| 3 | NBD·Pi ⇌ NBD + Pi | 260 ± 20 | 5 ± 1 | 1.3 ± 0.2 | |
| 4 | NBD·ADP·Pi ⇌ NBD·ADP + Pi | 235 ± 50 | 0.20 ± 0.06 | 0.047 ± 0.01 | |
| 5 | NBD·ADP·Pi ⇌ NBD·Pi + ADP | 0.70 ± 0.25 | 27 ± 6 | 0.019 ± 0.005 | |
| cat | NBD·ATP → NBD·ADP·Pi | | | | 0.065 ± 0.01 |

measures ADP lifetimes by ADP displacement[44]. The assay relies on the fluorescently labelled ADP derivate, N8-(4-N'-methylan-thraniloylami-nobutyl)-8-aminoadenosine 5'-diphosphate (MABA-ADP), that shows a substantial increase of fluorescence by 140% upon binding to

nucleotide-free Hsp70[44] (Supplementary Fig. 12a). Nucleotide release is then determined by real-time fluorescence measurements of Hsp70 with bound MABA-ADP in the presence of an excess of non-fluorescent ADP. The non-fluorescent ADP outcompetes re-binding of MABA-ADP

such that mono-exponential MABA-ADP unbinding is directly observed. We used this assay to determine $k^2_{off\ (MABA-ADP)}$ = 0.052 ± 0.01 s$^{-1}$ and $k^5_{off\ (MABA-ADP)}$ = 0.0064 ± 0.001 s$^{-1}$ (Supplementary Fig. 12b, c). These values correspond very well to those determined as part of the functional cycle, $k^2_{off\ (MABA-ADP)}$ = 0.051 ± 0.008 s$^{-1}$ and $k^5_{off\ (MABA-ADP)}$ = 0.006 ± 0.008 s$^{-1}$ (Supplementary Table 2, Supplementary Fig. 12d–f, Supplementary Fig. 13). Similarly, the $k^1_{off\ (MABA-ATP)}$ = 0.035 ± 0.008 s$^{-1}$ is in very good agreement with a value reported in the literature[66] $k^1_{off\ (MABA-ATP)}$ = 0.032 s$^{-1}$. Notably, the in-cyclo parameters for ADP ($k^2_{off\ (ADP)}$ = 0.11 ± 0.04 s$^{-1}$ and $k^5_{off\ (ADP)}$ = 0.019 ± 0.005 s$^{-1}$) are different from those of MABA-ADP by only a factor of ~2. This effect presumably arises from the hydrophobic MABA group that slightly enhances the affinity to the NBD. Altogether, the comparison of our kinetic parameters with independent experiments shows that the in-cyclo experiment faithfully reports the ADP and ATP lifetimes of the system.

## Consequences of variable metabolite concentrations on the BiP NBD functional cycle

With the complete set of kinetic constants at hand, we set out to simulate how physiologically relevant changes in ATP, ADP and phosphate concentration affect the functional cycle. In healthy cells, the ADP:ATP ratio is typically considered to be 1:10, and the ADP concentration in the ER is considered to increase up to 3 mM[67–69]. The free phosphate concentration is reported to vary between 0.5 and 5 mM[70,71]. We thus set the ATP concentration to 10 mM and probed variable ADP concentrations from 1 to 3 mM and phosphate concentrations from 0.25 to 5 mM (Supplementary Fig. 14). We simulated the relative populations of the states and their average occupancy time. Importantly, whereas in the ADP- and phosphate-free cycle, the average occupancy times of the NBD·Pi and NBD·ADP states result solely from dissociation of the ADP·Pi-bound state, under the cellular conditions they include contributions from ADP and Pi re-binding. Such re-binding events thus act as a buffer element to slow down the functional cycle.

Starting from physiological conditions of [ATP] = 10 mM and [ADP] = 1 mM, an increase in phosphate concentration from 0.25 to 5 mM resulted in an increased occupancy time of the NBD·ADP·Pi state from 34 to 240 s, whereas the occupancy time of the NBD·ADP state remained constant at ~10 s. Large phosphate concentrations thus prolong the functional cycle by acting as a buffer against dissociation of the NBD·ADP·Pi state. The same is true for increased ADP concentrations, which prolong the NBD·ADP·Pi occupancy time through ADP rebinding. Under conditions of both increased ADP concentration of 3 mM and phosphate concentration of 5 mM, the increase of the NBD·ADP·Pi occupancy time was 18-fold in comparison to the in-cyclo experiment, and the overall cycle was accordingly 8-fold longer. These numbers are in excellent agreement with the single batch experiments from pure ATP without the ATP regeneration and phosphate removal systems (Supplementary Fig. 10d). The kinetic parameters of the functional cycle thus give access to quantification of the functional cycle regulation under a wide range of physiologically relevant conditions. In the context of the full-length Hsp70 protein, such regulation could be a key factor to modulate client contact time in response to different cellular conditions.

## Calcium does not affect the functional cycle

Calcium is a key ER stress marker and plays a fundamental role in regulating the activity of multiple ER proteins[50]. While the role of magnesium ions in the catalytic activity of Hsp70 is well established[57], the potential role played by calcium in the Hsp70 functional cycle is not yet well understood[72]. This question is of special interest for the BiP as the ER can show large variations in calcium concentrations, during protein folding homeostasis and stress, with concentrations ranging from 1 mM under homeostatic conditions up to 0.1 mM under stress conditions[73,74]. It has been proposed that the calcium concentration might decrease the ATP hydrolysis rate of BiP by 2-fold at physiological calcium levels compared to no calcium[57,75] and increase the MABA-ADP-bound lifetime in a concentration-dependent manner by up to 4-fold[72]. Published crystal structures of the nucleotide-bound BiP NBD in the presence of calcium show that it binds in the nucleotide-binding site[72], in which the ADP-bound state calcium contacts both phosphate groups (α and β) while magnesium only contacts the β phosphate (Supplementary Fig. 15a). This might suggest a mechanism for the variation of the kinetic parameters, if Ca$^{2+}$ replaces Mg$^{2+}$ ions. Importantly, the calcium concentration is always lower than the magnesium concentration, also in the ER[76,77]. Accordingly, to test the effect of calcium on the kinetic parameters of the BiP NBD in the following experiments, we keep the magnesium concentration at 10 mM and vary the calcium concentration as indicated for each experiment.

First, we probed calcium binding to ADP-bound NBD by NMR equilibrium experiments. Upon addition of 3.3 mM calcium in the presence of 10 mM magnesium, large CSPs were observed, clearly confirming that calcium binding occurred (Supplementary Fig. 15b, c). Mapping of these CSPs on the structure showed changes consistent with a calcium binding site in lobes IA and IIA as expected from the published crystal structure of NBD bound to ADP·Ca$^{2+}$ (PDB 6ZYH)[72] (Supplementary Fig. 15d). Next, we used MABA-ADP release to measure the ADP mean life as a function of calcium concentration, at a fixed concentration of magnesium, showing an IC$_{50}$ of 212 ± 7 µM, in excellent agreement with literature[72] (Supplementary Fig. 16). Surprisingly, in the presence of 0.5 mM phosphate, the lifetime of bound ADP·Pi is completely independent of the calcium concentration (Supplementary Fig. 16).

Since the functional cycle is dominated by the NBD·ADP·Pi-bound state, the data readily suggest that the presence of calcium should have no effect on the functional cycle. Indeed, in the presence of 3.3 mM calcium the molecular hydrolysis rate in-cyclo of $k_{hydr}$ = 1.43 ± 0.06 min$^{-1}$ is essentially identical to the one in the absence of calcium ($k_{hydr}$ = 1.47 ± 0.05 min$^{-1}$) (Fig. 5a). Also, the relative populations of the three states were essentially the same, $t_T$ = 15 ± 2 s, $t_{DPi}$ = 19 ± 2 s and $t_D$ = 8 ± 2 s in presence of calcium, compared to $t_T$ = 15 ± 3 s, $t_{DPi}$ = 18 ± 2 s and $t_D$ = 8 ± 2 s in the absence of calcium (Fig. 5d and Supplementary Fig. 17). In full agreement with the unchanged kinetics parameters, no NMR signals for the NBD·ADP·Ca$^{2+}$ or NBD·Ca$^{2+}$ were observable excluding their significant formation within the functional cycle (Supplementary Fig. 17).

Finally, we wanted to explore the effect of calcium on conditions with variable ADP concentration that represent the conditions in the ER. We thus exploited our modular experimental setup in the absence of the ATP regeneration system, with and without phosphate, to generate conditions where the ADP concentration increases (Supplementary Fig. 18). We determined the ATP hydrolysis rate and the populations of the different states for different ADP:ATP ratios and phosphate concentrations. We observed NMR signals corresponding to the NBD·ADP·Ca$^{2+}$-bound state, but not the NBD·Ca$^{2+}$-bound state, with population and occupancy time increasing with the ADP concentration (Fig. 5e). When free phosphate was allowed to accumulate in parallel, the occupancy time of the NBD·ADP·Ca$^{2+}$ state decreased, showing that the phosphate prevents the calcium from binding (Fig. 5f).

Our results thus reveal an unexpected mode of action for calcium binding, in that calcium binds neither to the ATP-bound state, nor to the ADP·Pi-bound state resulting from ATP hydrolysis. Significant formation of the off-pathway ADP·Ca$^{2+}$-bound state is only relevant at high ADP and low Pi concentrations. This creates an interesting mechanism by which only the combined increase in ADP and calcium concentrations, at low Pi concentration, leads to the formation of the ADP·Ca$^{2+}$-bound state which lengthens the functional cycle.

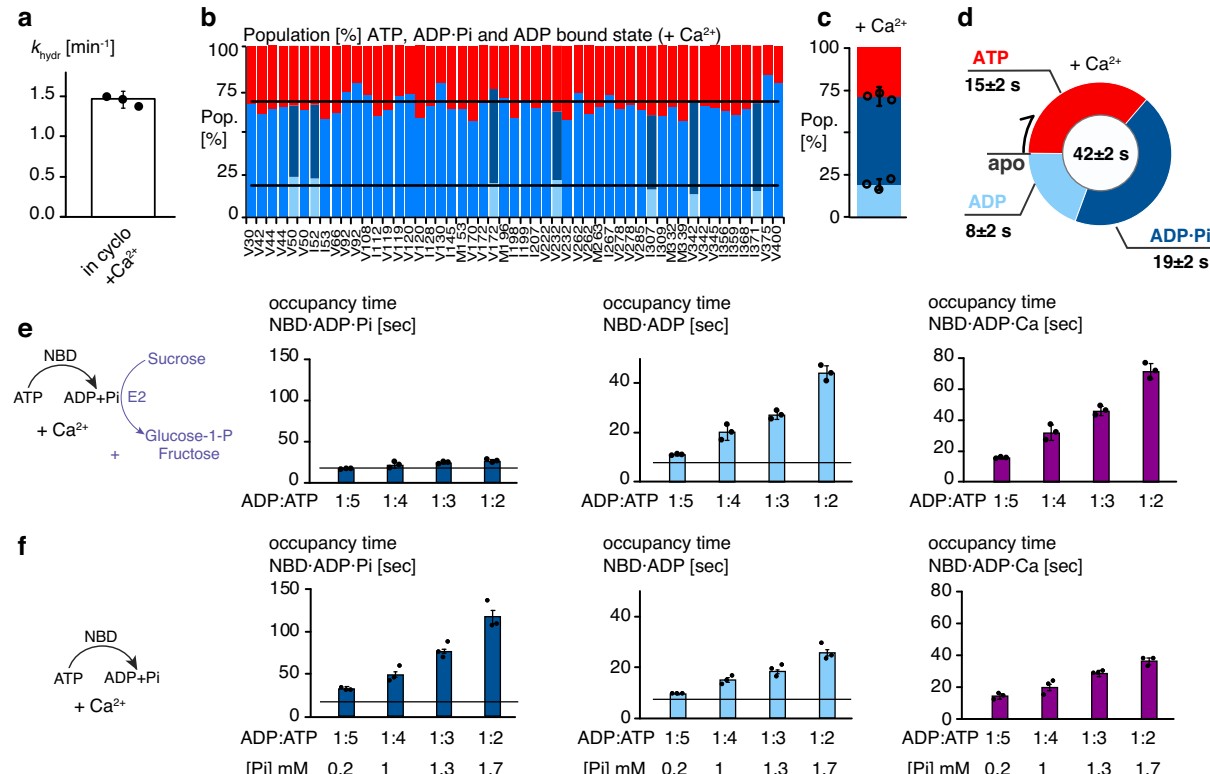

**Fig. 5 | Calcium does not affect the NBD functional cycle. a** Molecular hydrolysis rate in the presence of 3.3 mM calcium ([Mg$^{2+}$]/[Ca$^{2+}$]=3). Data points represent three independent experiments. The bar represents mean and standard deviation. **b** Residue-specific populations of the ADP (light-blue), ADP·Pi (blue), and ATP-bound state (red) in the presence of 3.3 mM calcium ([Mg$^{2+}$]/[Ca$^{2+}$]=3). Each bar corresponds to one methyl group. **c** Average populations of the ADP (light-blue) ADP·Pi (blue), and ATP-bound states. Data points represent three independent experiments. The bar represents mean and standard deviation. **d** Average length of

the NBD functional cycle including average state occupancy times in the presence of 3.3 mM calcium ([Mg$^{2+}$]/[Ca$^{2+}$]=3). **e, f** NBD average occupancy times of the ADP, ADP·Pi and ADP·Ca$^{2+}$-bound states at variable ADP:ATP ratio with E2 (**e**) and without E2 (**f**) in the presence of 3.3 mM calcium ([Mg$^{2+}$]/[Ca$^{2+}$]=3). Data points represent three independent experiments. The bar represents mean and standard deviation. The average occupancy times of the same experiment in the absence of calcium is shown by a horizontal bar.

## Discussion

The experimental in-cyclo NMR approach presented in this work provides a unique setup that allows atomic resolution observations of a molecular machine during its functional cycle, and provides access to the kinetic parameters. For the NBD of BiP, the method revealed kinetic parameters previously inaccessible, and thus for the first time a complete set of all 11 microscopic rate constants, resulting in a complete quantitative model of the functional cycle (Fig. 6). In particular, we resolved how ADP·Pi is released via two parallel pathways, either Pi first or ADP first, and that these pathways are both used under physiological conditions. ADP and phosphate thus can play crucial roles at physiological concentrations in extending the lifetime of the NBD functional cycle, making ADP·Pi release the rate-limiting step of the functional cycle. Attempts to determine the kinetic parameters without the data from the in-cyclo experiment did not yield a converging solution.

Under conditions mimicking the cytoplasmic environment ([ADP] = 1 mM, [ATP] = 10 mM, [Pi] <1 mM), we observe only a mild, 3-fold increase of the ADP and ADP·Pi occupancy times. However, an increase of ADP and phosphate concentrations from this point, potentially representative of conditions in the endoplasmic reticulum, lead to a significant increase of ADP- and ADP·Pi-bound states and thus a prolongation of the functional cycle. This raises intriguing questions of what role such modulations by ADP or Pi concentrations might play in the functional regulation of Hsp70 BiP. At physiological calcium concentration, the NBD·ADP·Ca$^{2+}$ complex can only form when elevated ADP concentrations are available, creating an off-pathway state that

prolongs the functional cycle in response to the increase of the ADP concentration. Therefore, BiP NBD requires a combined increase in ADP and calcium concentrations and low phosphate concentrations, to form significant off-cycle states and this pattern thus distinguishes BiP from other ER chaperones (Grp94[78], Calreticulin[79] and Protein disulfide isomerase[80,81]) whose activities are directly and strongly influenced by calcium concentration.

Our work has focused on the BiP NBD as a benchmark for subsequent studies. It will be exciting to extend our approach in a next step to full-length Hsp70 proteins, such as BiP, where we expect to understand the interplay between the substrate binding domain and the NBD. Similarly, our experimental setup might provide key information on how post-translational modifications, such as ampylation in BiP[82,83], modify the functional cycle timing. Additional interesting implications will be to understand how co-chaperones modulate the functional cycle. Nucleotide exchange factors[84,85] and nucleotide exchange inhibitors[86,87] accelerate and inhibit ADP release, respectively, and it will be interesting to study their effect in the context of a full-length Hsp70. Our in-cyclo setup provides a unique method to resolve how they target the two-step sequential release of the Pi and ADP and how their function might be connected to off-cycle pathways. Our experimental setup will be key to provide high resolution mapping of the interaction interface while simultaneously resolving the precise kinetics of the nucleotide conversion and release. Similarly, the methods described here will allow us to probe the effect of J-domain proteins on ATP hydrolysis with unprecedented precision. On the longer run, it will be interesting to see how the diverse cellular

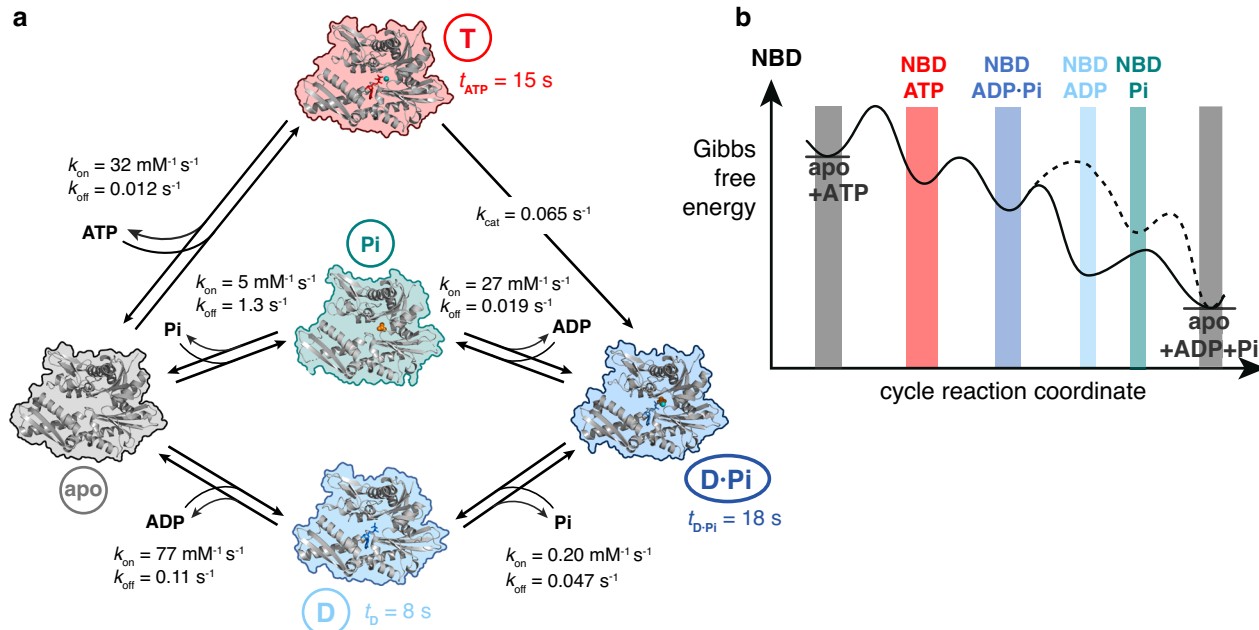

**Fig. 6 | The five-step functional cycle of BiP NBD and its energy landscape. a** The Hsp70 chaperone BiP NBD cycles through five states (apo state (PDB 3LDN), ATP-bound state (PDB 3LDL), ADP·Pi-bound state (PDB 5EVZ) Pi-bound state (PDB 3QFP) and ADP-bound state (PDB 7A4V)). The kinetic parameters were established in this work. **b** Schematic free energy landscape of the functional cycle. From the ADP·Pi-bound state, the NBD releases in a first step one of the molecules, resulting in either the Pi-bound (dashed line) or the ADP-bound state (plain line). From there, the apo state results by release of the second molecule.

functionalities of Hsp70 emerge from coupling ATP hydrolysis to interaction dynamics of the substrate binding domain.

## Methods
### Theory

The reaction space of the NBD is populated by the reaction scheme shown in Fig. 2a. It comprises 5 different states that are connected by 11 reaction equations of first and second order. We consider ATP hydrolysis an irreversible reaction without backreaction. The other pathways are reversible dissociation/association reactions, arbitrarily labeled 1–5. The microscopic equations (elementary reactions) for these 11 reactions are given by:

$$NBD^{ATP} \xrightarrow{k_{cat}} NBD^{ADP \cdot Pi} \tag{1}$$

$$NBD^{ATP} \underset{k^1_{on}}{\overset{k^1_{off}}{\rightleftharpoons}} NBD^{apo} + ATP \tag{2}$$

$$NBD^{ADP} \underset{k^2_{on}}{\overset{k^2_{off}}{\rightleftharpoons}} NBD^{apo} + ADP \tag{3}$$

$$NBD^{Pi} \underset{k^3_{on}}{\overset{k^3_{off}}{\rightleftharpoons}} NBD^{apo} + Pi \tag{4}$$

$$NBD^{ADP \cdot Pi} \underset{k^4_{on}}{\overset{k^4_{off}}{\rightleftharpoons}} NBD^{ADP} + Pi \tag{5}$$

$$NBD^{ADP \cdot Pi} \underset{k^5_{on}}{\overset{k^5_{off}}{\rightleftharpoons}} NBD^{Pi} + ADP \tag{6}$$

With the forward and backward kinetic rate constants connected by thermodynamic equilibrium dissociation constants given by

$$\frac{k^i_{off}}{k^i_{on}} = K^i_D \text{ for } i = 1, \dots, 5 \tag{7}$$

For each of the 5 states, a differential equation describing the conservation of mass can be derived:

$$\frac{d}{dt}\left[NBD^{apo}\right] = + k^1_{off}\left[NBD^{ATP}\right] + k^2_{off}\left[NBD^{ADP}\right] + k^3_{off}\left[NBD^{Pi}\right] \\ - k^1_{on}\left[NBD^{apo}\right][ATP] - k^2_{on}\left[NBD^{apo}\right][ADP] - k^3_{on}\left[NBD^{apo}\right][Pi] \tag{8}$$

$$\frac{d}{dt}[NBD^{ATP}] = + k^1_{on}\left[NBD^{apo}\right][ATP] - k^1_{off}\left[NBD^{ATP}\right] - k_{cat}\left[NBD^{ATP}\right] \tag{9}$$

$$\frac{d}{dt}\left[NBD^{ADP \cdot Pi}\right] = + k_{cat}\left[NBD^{ATP}\right] + k^4_{on}\left[NBD^{ADP}\right][Pi] + k^5_{on}\left[NBD^{Pi}\right] \\ [ADP] - k^4_{off}\left[NBD^{ADP \cdot Pi}\right] - k^5_{off}\left[NBD^{ADP \cdot Pi}\right] \tag{10}$$

$$\frac{d}{dt}\left[NBD^{ADP}\right] = + k^4_{off}\left[NBD^{ADP \cdot Pi}\right] + k^2_{on}\left[NBD^{apo}\right][ADP] - k^2_{off} \\ \left[NBD^{ADP}\right] - k^4_{on}\left[NBD^{ADP}\right][Pi] \tag{11}$$

$$\frac{d}{dt}\left[NBD^{Pi}\right] = +k_{off}^5\left[NBD^{ADP\cdot Pi}\right] + k_{on}^3\left[NBD^{apo}\right][Pi] - k_{off}^3\left[NBD^{Pi}\right]$$
$$- k_{on}^5\left[NBD^{Pi}\right][ADP]$$
(12)

Furthermore, we have the total amount

$$\left[NBD^{apo}\right] + \left[NBD^{ATP}\right] + \left[NBD^{ADP}\right] + \left[NBD^{ADP\cdot Pi}\right] + \left[NBD^{Pi}\right] = [NBD]_0$$
(13)

These equations can be used for numerical fitting of steady-state and non-steady-state experimental data.

## In-cyclo condition

Under the in-cyclo conditions, by the action of the enzymes E1 and E2, the ADP and Pi concentrations are effectively set to zero. Furthermore, a steady state sets in and the time derivatives in the differential equations above can be set to zero. Together, this results in a subset of equations with simplified terms

$$0 = +k_{off}^1\left[NBD^{ATP}\right] + k_{off}^2\left[NBD^{ADP}\right] + k_{off}^3\left[NBD^{Pi}\right] - k_{on}^1\left[NBD^{apo}\right][ATP]$$
(14)

$$0 = +k_{on}^1\left[NBD^{apo}\right][ATP] - k_{off}^1\left[NBD^{ATP}\right] - k_{cat}\left[NBD^{ATP}\right]$$
(15)

$$0 = +k_{cat}\left[NBD^{ATP}\right] - k_{off}^4\left[NBD^{ADP\cdot Pi}\right] - k_{off}^5\left[NBD^{ADP\cdot Pi}\right]$$
(16)

$$0 = +k_{off}^4\left[NBD^{ADP\cdot Pi}\right] - k_{off}^2\left[NBD^{ADP}\right]$$
(17)

$$0 = +k_{off}^5\left[NBD^{ADP\cdot Pi}\right] - k_{off}^3\left[NBD^{Pi}\right]$$
(18)

The macroscopic hydrolysis rate $k_{hydr}$ is experimentally determined from monitoring the pyruvate production. From there, the reaction flux $f$ of the functional cycle results as

$$f = k_{hydr}[NBD]_0$$
(19)

The inverse of the hydrolysis rate is the **average length of the functional cycle** $T$.

$$T = k_{hydr}^{-1}$$
(20)

The flux is maintained along the cycle, resulting in the equations

$$f = k_{cat}\left[NBD^{ATP}\right]$$
(21)

$$f = \left(k_{off}^4 + k_{off}^5\right)\cdot\left[NBD^{ADP\cdot Pi}\right]$$
(22)

$$f = k_{off}^2\left[NBD^{ADP}\right] + k_{off}^3\left[NBD^{Pi}\right]$$
(23)

$$f = k_{on}^1\left[NBD^{apo}\right][ATP] - k_{off}^1\left[NBD^{ATP}\right]$$
(24)

$$k_{off}^2\left[NBD^{ADP}\right] = k_{off}^4\left[NBD^{ADP\cdot Pi}\right]$$
(25)

The concentrations of the individual ligand-bound states are determined experimentally by peak integration. Then, from Eq. (21), $k_{cat}$ is obtained. From Eq. (22), $\left(k_{off}^4 + k_{off}^5\right)$ is obtained. Equations (23) and (24) are not further productive, because $\left[NBD^{Pi}\right]$ and $\left[NBD^{apo}\right]$ are effectively below the detection limit.

For each state $i \in \left\{NBD^{apo}; NBD^{ADP}; NBD^{ADP\cdot Pi}; NBD^{ATP}; NBD^{Pi}\right\}$, the **average occupancy time** can be defined, the average time an NBD molecule spends in a state when going once through the cycle.

$$t_i = \frac{[i]}{f}$$
(26)

with

$$\sum_i t_i = T$$
(27)

The equations follow

$$t_{NBD\cdot ATP} = \frac{1}{k_{cat}}$$
(28)

$$t_{NBD\cdot ADP\cdot Pi} = \left(k_{off}^4 + k_{off}^5\right)^{-1}$$
(29)

From the reaction schemes follow also the **kinetic lifetimes** for each state, which are defined by the respective off reactions:

$$\tau_{NBD\cdot ATP} = \left(k_{cat} + k_{off}^1\right)^{-1}$$
(30)

$$\tau_{NBD\cdot ADP\cdot Pi} = \left(k_{off}^4 + k_{off}^5\right)^{-1}$$
(31)

$$\tau_{NBD\cdot ADP} = \left(k_{off}^2\right)^{-1}$$
(32)

$$\tau_{NBD\cdot Pi} = \left(k_{off}^3\right)^{-1}$$
(33)

$$\tau_{NBD\cdot apo} = \left(k_{on}^1[ATP]\right)^{-1}$$
(34)

Some of the average occupancy times are similar and some are identical to the kinetic lifetimes, depending on the local branching and reversibility of the kinetic network.

## Single turnover reaction

The single turnover experiment starts from a batch of stoichiometrically assembled $NBD^{ATP}$ complex. ATP hydrolysis and ATP dissociation occur on parallel kinetic pathways, according to Eqs. (1) and (2). At very low protein concentrations $[NBD^{ATP}]\ll K_D^1$, rebinding of ATP after dissociation, as well as the buildup of the products ADP and Pi, can be neglected and the observed ATP decay follows the analytical equation

$$[ATP]/[ATP]_0 = \frac{k_{off}^1}{k_{cat}^*} + \frac{k_{cat}}{k_{cat}^*}e^{-k_{cat}^* t}$$
(35)

with

$$k_{cat}^* = k_{cat} + k_{off}^1$$
(36)

At proteins concentrations $[NBD^{ATP}] \gtrsim K_D^1$, as it was used in our experiments, ATP rebinding after dissociation cannot be neglected, which is competitive with the product ADP that builds up over time. An effective rate constant $k_{cat}^*$ is observed with

$$k_{cat} \leq k_{cat}^* \leq k_{cat} + k_{off}^1 \qquad (37)$$

which is under our experimental conditions, closely reaching $k_{cat}$, i.e.

$$k_{cat}^* \approx k_{cat} \qquad (38)$$

## Cloning, expression and purification of human BiP NBD

Human BiP NBD lacking its ER signal sequence (residues 19-406) with an N-terminal His₆-Tag including a TEV cleavage sequence was generated by introducing a mutation (G407 to stop) using the QuikChange II mutagenesis protocol (Stratagene) into the Human BiP (residues 19-654) gene synthetized by Genscript. The gene was inserted through NcoI and XhoI into a pET28a expression vector (Novagen).

BL21-(DE3)-Lemo cells (New England Biolabs) were transformed and grown at 37 °C in M9 minimal medium prepared with 99.85% D2O (Sigma-Aldrich) containing $^{15}NH_4Cl$ (1 g/liter; Sigma-Aldrich) and D-glucose-d7 (2 g/L; Sigma-Aldrich). BiP NBD expression was induced at an OD₆₀₀ of 0.6 by adding 1 mM isopropyl-β-d-thiogalactopyranoside (IPTG; Sigma-Aldrich) and expression was continued at 25 °C for 12 hours. Cells were harvested by centrifugation at 5000 g for 20 min. The pellet was resuspended in 20 ml of lysis buffer per liter of culture (25 mM HEPES (Roth), pH 7.5, 150 mM NaCl (Sigma-Aldrich), 10 mM MgCl₂ (Sigma-Aldrich), 0.02 mg/mL ribonuclease (0.02 mg/mL; Sigma-Aldrich), deoxyribonuclease (0.01 mg/mL; Sigma-Aldrich) and phenylmethylsulfonyl fluoride PMSF (0.2 mg/mL; Sigma-Aldrich). Cell lysis was performed using a microfluidizer (Microfluidics) for three cycles at 4 °C. The soluble bacterial lysate was separated from cell debris and other components by centrifugation at 14000 g for 60 min and loaded onto a Ni-NTA column (Qiagen) equilibrated in buffer A (25 mM HEPES, pH 7.5, 150 mM NaCl, 10 mM MgCl₂). BiP eluted at 500 mM imidazole (Sigma-Aldrich) concentration and was dialyzed overnight against buffer (25 mM HEPES, pH 7.5, 300 mM NaCl, 10 mM MgCl₂). BiP NBD was denatured with 6 M urea and loaded onto a Ni-NTA column (Cytiva) equilibrated in buffer A + 8 M urea (Sigma-Aldrich). BiP NBD eluted at 500 mM imidazole concentration in buffer containing 8 M urea. BiP refolding was achieved by dialysis overnight at 4 °C against buffer (25 mM HEPES, pH 7.5, 300 mM NaCl, 10 mM MgCl₂). After refolding, the His-tag was cleaved by incubation with 1 mg of TEV per 50 mg of BiP in cleavage buffer (25 mM HEPES, pH 7.5, 300 mM NaCl, 10 mM MgCl₂, DTT 1 mM and EDTA 0.5 mM) overnight at 4 °C. BiP was separated from TEV and uncleaved BiP via a reverse Ni-NTA column (Cytiva) equilibrated in buffer A. BiP was then applied to an anion-exchange column (Cytiva) equilibrated in the buffer QA (25 mM Tris (Sigma-Aldrich), pH 8.5) and eluted with 250 mM of KCl. Finally, BiP NBD was concentrated by ultrafiltration and subjected to size exclusion chromatography (Superdex-200 16/600 PG; Cytiva) to further purify the proteins and adjust to its final buffer (25 mM HEPES, pH 7.5, 150 mM KCl and 20 mM MgCl₂). Afterward, BiP was concentrated by ultrafiltration and stored at −20°C until use. Final yield of purified protein was 15-20 mg for wild-type BiP NBD per liter of deuterated M9 minimal medium. The protein concentration was determined using: measurement of absorbance at 280 nm and the Pierce BCA Protein Assay (Sigma-Aldrich). The BiP NBD not refolded protein was purified according to the same protocol without the unfolding/refolding steps. The BiP NBD expressed in LB (Luria broth) medium was purified following the same purification protocol.

## Methyl labeling of human BiP NBD

Methyl-labeled BiP proteins were obtained by growing the expression cells in M9 minimal media prepared with 99.85% D2O (Sigma-Aldrich) containing $^{15}NH_4Cl$ (1 g/L; Sigma-Aldrich) and D-glucose-d₇ (2 g/L; Sigma-Aldrich). When the optical density (OD) at 600 nm reached 0.8, a solution containing the labeled precursors was added. For [U-²H, ¹⁵N, ¹²C], Met-[¹³CH₃]ε, Val-[¹³CH₃/¹³CH₃]γ1/γ2, Ile-[¹³CH₃]δ1: 100 mg of 2-Keto-3-(methyl-13C)-butyric-4-¹³C acid sodium salt (Sigma-Aldrich), 30 mg of L-leucine-d₁₀ (Sigma-Aldrich), 80 mg of α-ketobutyric acid methyl ¹³C (99%) 3,3-D2 (98%) (Sigma-Aldrich) and 100 mg of [¹³C]ε-L-methionine (Sigma-Aldrich). For [U-²H, ¹⁵N, ¹²C], Met-[¹³CH₃]ε, Val-[¹³CH₃/¹²C²H₃]γ1/γ2, Ile-[¹³CH₃]δ1: 100 mg of 2-Keto-3-(methyl-d3)-butyric acid-4-¹³C,3-d sodium salt (Sigma-Aldrich), 30 mg of L-leucine-d₁₀ (Sigma-Aldrich), 80 mg of α-ketobutyric acid methyl ¹³C (99%) 3,3-D2 (98%) (Sigma-Aldrich) and 100 mg of [¹³C]ε-L-methionine (Sigma-Aldrich). For [U-²H, ¹⁵N, ¹²C], Met-[¹³CH₃]ε, Val-[¹³CH₃]γ2, Ile-[¹³CH₃]δ1: 240 mg of 2-hydroxy-2-[¹³C]methyl-3-oxo-4,4,4-tri-[²H]-butanoate (pro-S acetolactate-¹³C, NMR-Bio), 30 mg of L-leucine-d₁₀, 80 mg of α-ketobutyric acid methyl ¹³C (99%) 3,3-D2 (98%) (Sigma-Aldrich) and 100 mg of [¹³C]ε-L-methionine (Sigma-Aldrich). One hour after the addition of the precursors, BiP expression was induced by adding 1 mM isopropyl-β-d-thiogalactopyranoside (IPTG; Sigma-Aldrich) at 25 °C for 12 hours.

## Mutagenesis, expression and purification of the BiP assignment mutants

The QuikChange II mutagenesis protocol (Stratagene) was used to introduce the point mutations in the BiP plasmid: I33V, I76V, I145V, M148L, M153L, I190V, M196L, I199V, I207V, M263L, M332L, M339L and V400I. Polymerase chain reaction primers were obtained from Microsynth. The expression and purification of the mutant proteins was performed as described for the wild-type proteins. The final yield of purified mutants was similar to wild-type, except for the mutant M148L that expresses at a very low yield and thus could not be analyzed.

## NMR spectroscopy

All NMR experiments for BiP NBD were performed in NMR buffer (25 mM HEPES, pH 7.5, 150 mM KCl and 20 mM MgCl₂) at 25 °C or 37 °C. The experiments were recorded on Bruker AscendII 700 MHz, Avance 800 MHz or Avance 900 MHz spectrometers running Topspin 3.0 and equipped with a cryogenically cooled triple-resonance probe. NMR data were processed with nmrPipe[88] and ccpnmr 3.0.4[89].

## Assignment of BiP NBD Met[CH₃]ε, Val[CH₃]γ1/γ2 and Ile[CH₃]δ1 methyl groups

Met[CH₃]ε, Val[CH₃]γ1/γ2 and Ile[CH₃]δ1 assignment was obtained using a structure-based approach combining mutagenesis and 3D ¹³C, ¹³C-resolved [¹H, ¹H]-NOESY experiments. The following point mutation were used: I33V, I76V, I145V, M148L, M153L, I190V, M196L, I199V, I207V, M263L, M332L, M339L, V400I. Each mutant sample was recorded at 37 °C with an adjusted duration depending on their final concentration (experimental time ranging from 120 to 240 min per sample). For each mutant, spectra were recorded for the ADP·Pi-bound state (5 mM) and in-cyclo experiment. Analysis and comparison of the library of mutant spectra allowed the assignment of 1/35 valines (3%), 6/25 isoleucine (24%) and 5/6 methionines (83%). The mutant M148L expressed at very low yield and was therefore assigned by process of elimination to the only methionine signal that was missing an assignment. The network of assigned residues was used to expand the assignment using 3D ¹³C, ¹³C-resolved [¹H, ¹H]-NOESY experiments recorded with sample concentrations ranging from 0.4 to 0.8 mM with a mixing time of 400 ms and an adjusted duration depending on the BiP NBD final concentration (experimental time ranging from 3 to 5 days per sample). Stereospecific assignment of the valine cγ1/cγ2 for residues located in the BiP NBD has been obtained using a sample

[U-$^2$H, $^{15}$N, $^{12}$C], Met-[$^{13}$CH$_3$]$^\varepsilon$, Val-[$^{13}$CH$_3$/$^{13}$CH$_3$]$^{\gamma1/\gamma2}$, Ile-[$^{13}$CH$_3$]$^{\delta1}$ and a 3D $^{13}$C, $^{13}$C-resolved [$^1$H, $^1$H]-NOESY experiments recorded with a short mixing time of 50 ms that directly correlate the two methyl groups within one amino-acid. All 3D NOESY experiments were fully sampled, i.e., no non-uniform sampling was used. Analysis of the 3D $^{13}$C, $^{13}$C-resolved [$^1$H, $^1$H]-NOESY cross peaks with the structure of the BiP NBD in the ADP·Pi state allowed to expand our mutagenesis-based assignment to assign 32/35 valines, 22/25 isoleucine and 6/6 methionine. Among the 32 valines, 24 had both methyl Cγ1 and Cγ2 assigned and 8 had one of them, resulting in a total of 84 observable NMR signals. The missing assignments correspond to methyl groups for which no NOESY network could be resolved.

## Exchange Spectroscopy (EXSY) experiments

$^{13}$C-edited 2D Exchange Spectroscopy (EXSY) experiments were acquired to characterize the exchange between the NBD apo and NDB·Pi state ($k_{3on}$, $k_{3off}$). For this purpose, we used a modified methyl-TROSY experiment with an additional EXSY element prior to acquisition (90° $^1$H – EXSY mixing delay – 90° $^1$H)[49]. A recycle delay of 1 s was used in all experiments. The EXSY mixing times were set to 5, 50, 100, 200, 300, 400, 500, 600, 800, 1000, 1250, 1500 ms, with an average acquisition time of 120 min for each spectrum. The EXSY experiments were performed at 25 °C with a [U-$^2$H, $^{15}$N, $^{12}$C], Met-[$^{13}$CH$_3$]$^\varepsilon$ and Ile-[$^{13}$CH$_3$]$^{\delta1}$ BiP NBD sample at 400 μM in NMR buffer (25 mM HEPES, pH 7.5, 150 mM KCl and 20 mM MgCl$_2$) with 200 μM of phosphate.

## Preparation of ADP-containing samples

All equilibrium NMR experiments were recorded in NMR buffer at a BiP NBD concentration of 100 μM. For the equilibrium experiments in the presence of ADP, 1 mM of ultrapure ADP (Sigma-Aldrich) was added and the NMR spectrum measured immediately afterwards to minimize ADP auto-hydrolysis. The same procedure was implemented for all NMR titrations experiments involving ADP in order to avoid the accumulation of free phosphate. For the NBD·ADP·Pi sample, ultrapure ADP (Sigma-Aldrich) and Pi were added to the sample at concentration of 5 mM. 1D $^1$H NMR spectra were used to monitor the purity of ADP and absence of contaminants.

## NMR in-cyclo experimental setup: Composition of the in-cyclo NMR samples

All in-cyclo NMR experiments were recorded with BiP NBD samples [U-$^2$H, $^{15}$N, $^{12}$C], Met-[$^{13}$CH$_3$]$^\varepsilon$, Val-[$^{13}$CH$_3$/$^{12}$C$^2$H$_3$]$^{\gamma1/\gamma2}$, Ile-[$^{13}$CH$_3$]$^{\delta1}$ in NMR buffer (25 mM HEPES, pH 7.5, 150 mM KCl and 10 mM MgCl$_2$) at a concentration of BiP NBD 100 μM. The in-cyclo ATP regeneration system was composed of 5 units of pyruvate kinase (E1) from rabbit muscle ($\approx$ 200 nM; Sigma-Aldrich) and the concentration of phosphoenolpyruvate (PEP- Sigma-Aldrich) was adjusted to maintain the system active for at least four hours ([PEP] = 10–100 mM). The in-cyclo phosphate removal system was composed of 5 units of recombinant sucrose phosphorylase (E2) ($\approx$ 500 nM; Sigma-Aldrich) and the concentration of sucrose (Sigma-Aldrich) was adjusted to maintain the system active for at least four hours ([Sucrose] = 10–100 mM). This experimental setup was modified as described in the text by selectively including E1 and/or E2 depending on the purpose of the experiments. All the experiments were recorded at minimum as experimental triplicates.

## Recording of the in-cyclo NMR experiments

The in-cyclo experiment was initiated by the addition of 5 mM ATP (Sigma-Aldrich) and allowed to equilibrate at a temperature of 37 °C for 5 min in the NMR spectrometer. After the equilibration time 2D [$^{13}$C,$^1$H]-HMQC spectra were recorded in interleaved mode with 1D $^1$H NMR spectra. The 1D $^1$H NMR spectra were used to monitor the pyruvate increase, the glucose-1-phosphate and the concentration of ATP using the NMR signal of 0.5 mM sodium 2,2-dimethyl- 2-silapentane-5-

sulfonate-d6 (d, 98%) (DSS; Cambridge Isotope Laboratories Inc.) as an internal reference. The same setup was used for the alternative experiments by including or excluding E1 and/or E2. When E1 was not included, the ratio of the ATP/ADP NMR signals was used to monitor their relative concentrations. For the NMR experiment with MABA-ATP (8-[(4-Amino)butyl]-amino-ATP-MAN; Jena-Bioscience), we found that pyruvate kinase was not able to efficiently process MABA-ADP and therefore the experiment was run in the absence of E1 and E2.

## MABA-ADP release assay

Nucleotide release and binding measurements were performed using the fluorescent nucleotide analogues MABA-ADP (8-[(4-Amino)butyl]-amino-ADP-MAN; Jena-bioscience) carrying a MANT moiety whose fluorescence increases upon binding to Hsp70s[44,55]. To measure nucleotide release, 1.8 μM fluorescent nucleotide analogues were incubated with 1.8 μM proteins in NMR buffer for at least 45 min at 25 °C to allow for complex formation (Solution A). Solution B is NMR buffer with 7.5 mM ADP. The final concentrations after mixing were 1.44 μM for each protein, 1.44 μM fluorescent nucleotides, and 1.5 mM ADP. The measurements were started by adding 20 μL of solution B to 80 μl of solution A in a 96-well plate (Corning) and fluorescence (excitation 360 nm, emission 420 nm) was detected over time with a fluorescence spectrometer (Tecan Plate Reader) at 37 °C. The dead time between sample mixing and signal recording was ~ 2 s. All solutions contained 25 mM HEPES, pH 7.5, 150 mM KCl, 10 mM MgCl$_2$ at 37 °C. The solutions A contained CaCl$_2$ and phosphate where indicated (final concentrations are stated in figure legends). The dissociation rate constants ($k_{off}$) were determined by fitting the data to a one phase exponential function using a python script.

## ATP analog preparation

AMPPNP (adenylyl imidodiphosphate; Jena-bioscience), AMPPCP (adenylyl methylenediphosphate; Jena-bioscience) and ATP-γ-S (adenosine 5′-(gamma-thiotriphosphate; Jena-bioscience) were dissolved in the HKM reaction buffer (25 mM HEPES, pH 7.5, 150 mM KCl and 10 mM MgCl$_2$) at a concentration of 50 mM directly before to be mixed with the BiP NBD sample. Samples were prepared fresh before each use to minimize the known instability of these analogs that undergo slow hydrolysis in solution.

## NADH ATP assay

BiP NBD not-refolded or refolded was diluted to a final concentration of 1 μM in the HKM reaction buffer (25 mM HEPES pH 7.5, 150 mM KCl and 10 mM MgCl$_2$). The reaction buffer also contained 10 mM Phosphoenolpyruvate (Sigma-Aldrich), 0.5 mM NADH (Sigma-Aldrich), pyruvate kinase (Sigma-Aldrich) and lactate dehydrogenase (Sigma-Aldrich) (diluted to 5 U). The reactions were started by addition of 5 mM ATP in a final volume of 160 μl in 96-well microplates (Corning) and NADH absorbance at 340 nm (A$_{340}$) was measured over time at 37 °C with a Tecan Plate Reader. Linear regression analysis was performed and the ATP hydrolysis activity was calculated using the molar extinction coefficient of NADH ($\varepsilon$ = 6220 M$^{-1}$ cm$^{-1}$).

## Single turn-over assay

The ATPase activity of BiP NBD was determined under single-turnover conditions as previously described with minor modifications of the protocol[44]. The main modification was to replace the radioactivity-based detection by a label free anion-exchange based quantification[65]. The BiP NBD-ATP complex was formed by mixing BiP NBD with 10 mM ATP in formation buffer (25 mM HEPES pH 7.5, 50 mM KCl, 10 mM MgCl$_2$) and incubating the mixture for 5 min on ice. The BiP NBD was separated from unbound ATP at 4 °C by gel filtration on NICK columns (Cytiva) equilibrated in reaction buffer (25 mM HEPES pH 7.5, 150 mM KCl, 10 mM MgCl$_2$). Fractions containing the BiP NBD ATP complexes were pooled and snap frozen in liquid nitrogen and kept at −80 °C. For

the ATPase activity determination, BiP NBD ATP complex solutions were thawed and the reaction started by mixing with the reaction buffer prewarmed at 37 °C (25 mM HEPES pH 7.5, 150 mM KCl, 10 mM MgCl$_2$). At given time points, samples were withdrawn from the reaction, mixed with an equivalent volume of HCl and snap frozen in liquid nitrogen and kept at −80 °C. The relative ATP/ADP ratio was determined using an anion chromatography-based assay.

## ITC experiments

ITC data were collected with an MicroCal PEAQ-ITC Automated instrument (Malvern Panalytical) at 25 °C, 10 μcal/sec reference power, 500 r.p.m. stirring speed, and high gain. 18 injections of 2 μl of 250 μM ADP or 250 μM MABA-ADP were titrated at 250 sec intervals into 25 μM BiP NBD or 25 μM BiP NBD and 10 mM Pi. Experiments for MABA-ADP were also titrated into 50 μM BiP NBD or 50 μM BiP NBD and 10 mM Pi. Baseline correction, peak integration, and data fitting were conducted using the MicroCal PEAQ-ITC Analysis Software (v 1.41) and the initial injection was not included for data analysis. Data were fit to a single-site binding model to derive $K_D$, $\Delta H$ (change in binding enthalpy), and $N$ (stoichiometry) values from the binding isotherm.

## SEC-MALS

SEC-MALS measurements of BiP NBD were performed at 25 °C in NMR buffer (25 mM HEPES, pH 7.5, 150 mM KCl and 20 mM MgCl$_2$) using a SEC-3 HPLC column (300 Å pore size; Agilent Technologies) on an Agilent 1260 HPLC. Elution was monitored by multiangle light scattering (Heleos II 8 +; Wyatt Technology), differential refractive index (Optilab T-rEX; Wyatt Technology) and absorbance at 280 and 254 nm (1260 UV; Agilent Technologies). All the system parameters were calibrated using an injection of 2 mg/ml BSA solution (ThermoPierce) and standard protocols in ASTRA 6. Molar mass (MM) and mass distributions were calculated using the ASTRA 6 software (Wyatt Technology).

## Kinetic fitting

The experimental data of the BiP NBD in-cyclo experiments were fitted using standard reaction kinetics. Free variables were the $k_{on}$ and $k_{off}$ rates for the five reversible reaction and the catalytic rate constant ($k_{cat}$) for the irreversible ATP hydrolysis step (See theory section). The initial values for $k_{cat}$, $k_{off}^2$, $k_{off}^4$ and $k_{off}^5$ were set to the value determined experimentally with in-cyclo experiment. The initial value for $k_{off}^3$ was taken from the EXSY experiment and the initial range for $k_{off}^1$ and $K_D^1$ were taken from the literature[66]. From these starting conditions, all kinetic rates were inferred from Markov Chain Monte Carlo (MCMC) simulations, with the constraints that the ratios of $k_{off}$/ $k_{on} = K_D$ had to be in a range (± 50%) to their experimentally determined dissociation constants ($K_D$) by ITC ($K_D^2$ and $K_D^5$) or NMR ($K_D^3$ and $K_D^4$). The prior distributions were defined based on the average state occupancy resolved from the in-cyclo experiment: $t_{NBD·ADP}$, $t_{NBD}$, $t_{NBD·ADP·Pi}$ and $t_{NBD·Pi}$. Numerical integration of the corresponding differential equations for the reaction scheme (described in Fig. 2a) was performed using a custom Python module (Kinetic).

To fully determine all kinetic parameters, the data fitting procedure was conducted as a global fit of four independent experimental datasets: (i) the in-cyclo experiment with ATP regeneration and phosphate removal, (ii) ATP regeneration only, (iii) single batch experiments starting from pure ATP with the phosphate removal system, and (iv) single batch experiments starting from pure ATP.

## Reporting summary

Further information on research design is available in the Nature Portfolio Reporting Summary linked to this article.

## Data availability

All data needed to evaluate the conclusions of the paper are presented in the manuscript. PDB codes of previously published structures used in this study are: 5EVZ, 3LDL, 3LDN, 3QFP, 7A4V, 5F1X, 3LDO, 5FR2, 6DFM, 6DFO, 6DO2 and 6ZYH. The 84 state-specific methyl resonance assignments of BiP NBD have been submitted to the Biological Magnetic Resonance Data Bank under accession code 51751. The source data underlying Figs. 1c, 3c, 3e, f, 4e, f, 5a, 5c–f and Supplementary Figs. 1b, 2b, 4d–g, 9b, 10a–d, 11a, b, 12c–f, 13b, 14a, b, 16a, b are provided as a Source Data file. Source data are provided with this paper.

## Code availability

Kinetic a Python module collecting functions useful for simulation of chemical and enzymatic kinetics is available at https://git.scicore. unibas.ch/researchit/kinetic.

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

## Acknowledgements

This work was funded by the Swiss National Science Foundation (grants 185388 and 207755 to S.H.). The authors thank Anna Leder for critical reading of the manuscript and Tim Sharpe and Thomas Müntener for technical support. The ITC measurements were supported by SNSF R'Equip Grant 213436.

## Author contributions

G.M. conducted the experiments. G.M. and S.H. designed the study, analyzed the data, discussed the results, and wrote the manuscript.

## Competing interests

The authors declare no competing interests.
