## [Transparent Peer Review file · Nature Communications]

Mechanism of ATP hydrolysis in the Hsp70 BiP nucleotide-binding domain

Corresponding Author: Professor Sebastian Hiller

Version 0:

Reviewer comments:

Reviewer #1

(Remarks to the Author)

Mas and Hiller present a new approach to characterize ATPase domains in cyclo with high temporal and spatial resolution. By coupling advanced isotope labeling for methyl-TROSY, atomic-resolution fingerprints of domains in various nucleotide states, and real-time kinetics measurements, the authors gain insight into the functional cycle of an ATPase domain at unprecedented detail. The NMR approach is benchmarked on the Hsp70 nucleotide binding domain (NBD), with the model system being the human ER Hsp70 BiP. The authors compare their in cyclo method to conventional single-turnover ATPase assays and other fluorescence assays for nucleotide binding/dissociation, demonstrating that their in cyclo-NMR method matches the information content from the other assays while also providing atomic-resolution structural insight. Moreover, because the approach outlined by Mas and Hiller relies on an ATP re-generation system, they are able to study the NBD in a more "native-like" nucleotide state -- fluctuating between apo, ATP-bound, ADP-Pi-bound, and repeat. Their data convincingly show that conventional fluorescence-based ADP displacement assays overlook the important effect of the post-hydrolysis state, namely that Pi "locks" ADP into the NBD. This is a central advantage to the method outlined in this work: that the ATPase cycle can be studied without the need for fluorophores or displacement assays. Finally, the authors find no effect for Ca²⁺ on the BiP NBD cycle, suggesting that it remains buffered from Ca²⁺ flux that occurs during ER stress.

Overall, the paper is very clearly written, and the results are logically presented. The data supports the claims in the manuscript, and I recommend the manuscript for publication in Nature Communications following only minor changes. My main concern regards the NMR assignments for the ATP-bound state. I've outlined my concerns below, followed by other minor points.

1. Fig 3 – how were the assignments transferred from the ADP-Pi- to the ATP-bound state? The 13 mutants are mentioned, but this set includes only 1 Val residue. "Unambiguous spectral regions" assumes relatively isolated peaks; however, for the crowded Val region this is certainly not trivial (eg Supp Fig 6). Could the authors clarify how the assignments for the ATP-bound state were obtained, especially in the crowded Val region? Were NOEs in the ATP-bound state measured? Was the %T used as an assignment restraint?

2. Fig 3D – I agree that the average ADP-Pi % is ~75% over all resonances. But could the authors comment on why there is a spread in the % of the ADP-Pi-bound state of ~65% (V149) to ~90% (V92)? The two states are in slow exchange, so are differential ps-ns or us-ms dynamics within each state responsible for the differences?

Other points:

Results (Line 122) - 4 NOEs/methyl does not guarantee "unambiguous assignments as a network effect". This number is a reasonable value for an NOE-based methyl assignment, and fits with about the average number reported by Pritisanac et al. 2017 JACS. However, I don't think it's fair to say that 4 NOEs/methyl guarantees an unambiguous assignment, as even these automatic assignment methods based on advanced graph-theory or genetic algorithms cannot guarantee this.

Results (Line 122) – would the median number of NOEs/methyl be more relevant to report here? How many BiP methyls had fewer than 4 NOEs? How many did not have NOEs? (and if so how were the orphans assigned?)

Methods (Line 392) – At what temperature was BiP NBD refolded?

Methods (Line 424) – the mutants need clarification. In the early results, lines 115 and 126, it is mentioned that 6 Met were mutated followed by 7 mutations to residues in the core of the methyl-methyl NOE networks. But the Methods (in section line 424) state that one of the mutants failed to express (M148L) and could not be analyzed. So, I suppose then that only 5 of the 6 Met were assigned by mutagenesis and the last one was assigned by process of elimination?

Methods (Line 452) – I assume that the 3D NOESY experiments were fully sampled? (If not, how was the NUS performed and the sparse data reconstructed?)

Methods (Line 466) – the addition of co-chaperones and client proteins are mentioned here but I think this line can be removed entirely.

Methods (Line 524-5) – is there a reference needed for the ATP/ADP quantification by anion exchange?

Fig 1 legend (Line 743) – typo in “right” (righ)

Fig 1C – is “%” missing from right-axis label? It says “%” in the legend but not in figure.

Fig 1C – how do the authors define the % of number of observed NOESY contacts? (right-axis). I'm not sure what this value means. Eg for 8-10 Å the grey bar shows that ~25% of the NOEs based on the structure are actually observed, but what does the cross at ~90% mean?

Fig 3A, add red = ADP-Pi and black = ATP to the spectra for clarity. Especially because Fig 3B has a different spectral coloring scheme that is combined with the red/black text from 3A.

Figure 4A – the legend text is not clear what the data are in reference to (in cyclo or single turnover). It would be helpful to mention that this is a single turnover assay as described in Theyssen et al.

Fig 6A legend does not match the colors shown in the figure. I think 0.001 mM Ca²⁺ is also missing from the legend text.

Figure 6B add “no Pi” in purple text to the figure.

Fig 6B legend -- capital “D” for “Data points ...”

Fig 6C legend – delete “and black dash line”

Supp Fig 1c shows the full 2D 1H-13C HMQC of BiP. Why not show this in Figure 1 where assignments are used? Or show the Val region to demonstrate the completeness in assignment.

References to recent (1) real-time NMR work and (2) capturing of an ADP-Pi state on an ATPase could be added: PMID 32790300 and PMID 38326645

Line 70 – “dependants” -> depends

Line 79 – individual cycle steps

Line 80 – “ER” already defined

Line 83 – characterize the absence

Line 109 – in the presence of ADP

Line 377 – genscript needs capitalization

Reviewer #2

(Remarks to the Author)

Comments on Mas and Hiller, 2024

The work of Mas and Hiller study the catalytic cycle of ATP in an isolated NBD of human BiP. They use different techniques such as NMR and conventional activity assays (at steady state and single turnover) to understand the step of ATP hydrolysis, ADP-Pi release, etc. They also used different metals, such as calcium, to apply their study in a more physiological state. The main findings are that ADP-Pi release is the rate-limiting step in the functional cycling of BiP NBD, explaining why previous work has not seen the same effect between ATP analogues and ATP, and that neither Calcium nor free inorganic phosphate has a significant effect on its cycling. The paper is very straightforward and well written and I would like to suggest some improvements in the overall story.

I would recommend the authors to include a schematic representation of the Hsp70 cycle, it would be helpful to understand the dynamics and conformational changes of the protein during the functional cycle

At the end of the introduction, the authors describe the main conclusion of the work, for instance about the effect of calcium. However, the introduction provides very little information about the expected or known effects of calcium. I would recommend to add to include a short background about its reported effects

In the results section titled 'NMR Resonance Assignment of NBD Methyl Groups', the authors conducted various analyses to confirm the protein's native-like state after refolding. It is unclear from the text whether the protein's activity remained the same before and after the unfolding/refolding cycle.

Does BiP form a dimer with the NBD domain alone? BiP has been shown to dimerize (Rivera et al., 2023), and it is possible that some dimers could be formed over the NBD itself. Have you checked for the presence of an NBD-NBD dimer in your system?

As they mentioned they are using a highly purified BiP, I would recommend adding the % of purity in the SDS-PAGE gel.

The process of assembling the BiP-ADP-Pi complex is not sufficiently explained.

They verified their NMR assignment by single-point mutagenesis. Did the author confirm that these modifications do not affect the proper folding and activity of the BiP NBD?

Is it possible to use NMR data to depict the conformational transition of the BiP NBD domain? For example, in the statement "Additional large chemical shift differences were, however, also observed in lobe IA. These reflect the rotation of lobe IA resulting from ATP binding, which, in the full-length protein, results in docking of the SB" is it only possible to draw this conclusion using other types of structural data, such as cryo or x-ray? I recommend labeling the lobes in the 3D structures of the BiP NBD as shown in Figure 3C of the paper.

They state that "The NMR signal intensities of each of the two states is proportional to their relative population and thus to the kinetic rate constants that connect the two states" It would be helpful to include a more detailed explanation of this.

In the discussion, they say "Formation of the ADP-Pi post-hydrolysis state has major consequences for the regulation of the functional cycle in response to environmental change". However, this statement only applies when the NBD domain is alone, which is not the case in the cell. It would be interesting to discuss the possible effects of the SBD and the substrate. Additionally, it is unclear what the authors expect to observe. The study found that Ca does not have any effect on ADP-Pi lifetime. However, it is unclear whether this is due to the absence of the SBD. The discussion section could be improved.

The discussion section could be improved by correlating the different binding and movement of the NBD domain with specific amino acids in the structure. For instance, Vogel et al. (2006) demonstrated the significance of many amino acids for ATP and ADP-Pi binding, and it would be beneficial to discuss how they vary across different species. Additionally, the authors briefly mention the importance of their findings with the allosteric cycle of BiP, but further elaboration would enhance the discussion. A deeper explanation of the amino acids involved in allosteric communication is recommended to provide more clarity. Predictions could also be made based on this well-known information.

Post-translational modifications of BiP are not mentioned, and now it is an important line of research. I would recommend discussing how different post-translational modification or co-chaperones binding could affect BiP ATP cycle.

In lines 214 and 215 the authors mention non-canonical behavior of ATP analogs. This finding is interesting as it has been observed before but not clearly explained, both in vivo and in vitro. It is suggested to discuss this further in the discussion section and correlate it with other papers that show similar behavior (Flynn et al., 1989, Liberek et al., 1991, and from our group Rivera et al., 2023)

Minor comments

Line 51 In my understanding the abbreviation BiP is Immunoglobulin Binding Protein, that is why the i on BiP is lower case

Line 342 I suggest rephrase "hydrolysis rate determined in cyclo matches the value determined by single turn-over experiments"

Line 350 it say "call" instead of "cell"

Line 383 ml should be mL

Line 402 Which method did they use to measure protein concentration?

Line 497-498 and 503 "HKM reaction buffer HKM" seems redundant

Line 502 Instead of "NADH ATP assay" I suggest called, "Coupled ATP assay" or "NADH coupled ATP assay"

Line 505 there is an extra space after the first Sigma-Aldrich

Some of the reagents in Materials and method does not show the origin (as example IPTG)

Francesca Burgos-Bravo and Christian A.M. Wilson

Reviewer #3

(Remarks to the Author)

The authors present an NMR kinetics analysis of the BiP NBD domain under ATP-regeneration conditions previously described, allowing the study of the enzyme under steady-state conditions. This originality lies in the ability to monitor the visible states of the protein in action at its steady-state regime. They identified two states and assigned them to the ATP-bound and ADP.Pi-bound NBD states. The kinetic parameters align with previous measures, raising questions about the necessity of this method. Finally, they claim that the catalytic kinetics are unaffected by bulk phosphate and calcium, which contradicts prior studies. Moreover, an alternative interpretation of the data can be proposed, making the latter assertions speculative.

The method is interesting but lacks clear definitions of kinetic parameters and mathematical formalism, making the conclusions difficult to grasp. Some assertions lack evidence, and alternative explanations could align with the results. The method seems oversold, requiring more rigorous explanations to be persuasive. Additionally, the manuscript is riddled with numerous typos and grammatical errors that need correction.

Significant revisions are required for the manuscript. Beyond benchmarking the method, the authors do not address common aspects of the ATPase chaperone field, such as the effects of ATP hydrolysis enhancers, ADP exchange factors, or the analysis of various mutants to demonstrate important conformational changes for specific chemical transitions. I suggest that the authors rigorously define the physics and mathematics required for proper data analysis and include more control experiments to minimize ambiguous interpretations. Given that the main claim of this work is to benchmark the steady-state method for use in more complex systems, it might be better suited for a specialized journal focusing on method development.

Major Concerns:

1. Precise Definition of Experimental Conditions and Observables:

The manuscript lacks clear definitions of reaction schemes and mathematical frameworks essential for understanding the authors' reasoning and calculations. I recommend providing a reaction scheme, defining both microscopic (associated with single transitions) and macroscopic (apparent) rate constants, and outlining their relationships, possibly in supplementary material. The theoretical enzymology framework, crucial for supporting the authors' hypotheses, is inadequately addressed. Clarification is needed on the distinction between "in cyclo" experiments and steady-state experiments. If they overlap entirely, the rationale for different terminology should be explained.

In Figure 7b, the proposed energy diagram, focusing solely on the protein, mistakenly implies equal free energy for the two apo states, overlooking that it pertains to the entire system (enzyme + substrate). This should be corrected for accuracy and could help define various constants and transitions early in the manuscript.

p.7-l.159 needs a clearer mathematical and theoretical explanation for the calculations of $k(\text{hydr})$ and its relationship with $k(\text{cat}) + k(\text{off ADP})$.

p.9-l.226 requires a thorough mathematical analysis to accurately calculate microscopic rate constants, considering the lifetime of ATP-bound NBD and ADP-bound NBD and discussing states like the apo state that may not be visible due to line shape broadening.

2. Critical Evaluation of the Observables:

Questions arise about whether the calculated exchange rates for each transition are consistent with the chemical shift differences between states, indicating slow exchange processes. The absence of visible states in NMR signals, influenced by exchange rates, underscores the need for discussion on dynamic parameters' impact on NMR properties. Different experimental settings could have been employed to vary bulk ATP and ADP concentrations or use mutants to explore the steady-state regime under diverse conditions more thoroughly.

3. Misleading Statements on Atomic-Scale Structural Information:

CSP analysis provides low-resolution structural insights, influenced by multiple factors such as direct ligand binding or allosteric effects. This complexity challenges precise ligand positioning on a protein surface, rendering CSP analysis a lower-resolution method for this purpose.

4. Demonstrating the Nature of the Unknown State:

Beginning with NMR data on AMPPCP, AMPPNP, and ATPyS binding before presenting steady-state data would improve clarity. Defining an enzymatic dead mutant could have offered insights into the ATP-bound state, indirectly inferred. The identification of the unknown state, potentially a specific ATP-bound state or an alternative conformation, requires further investigation to clarify its nature during enzyme activity.

Additional issues:

The following are independent major concerns that need to be addressed:

p.9-l.232: "The ATP lifetime in cyclo is identical to classical single turn-over experiments."

This chapter presents several issues: Firstly, the statement "Conventional approaches do not allow determination of the ATP hydrolysis rate $k(\text{cat})$ from a steady-state experiment" is incorrect. It is possible to measure $k(\text{cat})$ using a proper setting of the initial conditions of an enzymatic experiment and by defining the reaction scheme, and then, fitting the measured initial velocities against the substrate concentrations. Secondly, comparing the inverse of the microscopic rate constant $\tau T = 13.4 \pm 2$ s (in cyclo) with the inverse of the macroscopic rate constant of ATP consumption in a single turnover reaction, $\tau T = 14.3 \pm 2$ s (single turnover), is invalid. The latter is based on an apparent rate constant influenced by various transitions and experimental conditions, while the former is specific to the hydrolysis step. To compare $k(\text{cat})$ from the NMR steady-state experiment to $k(\text{cat})$ obtained from an established enzyme assay, applying the Michaelis-Menten formalism is initially recommended before considering more complex reaction schemes if significant discrepancies arise between methods. p.10-l.275: "Because the ADP lifetime in equilibrium experiments is strongly correlated to the bulk phosphate concentration, ..."

Your experiment validates that the $K(\text{off})$ of NBD-bound ADP depends on bulk phosphate concentration, but this doesn't necessarily imply a change in lifetime. If $K(\text{on})$ also depends on P_i concentration in a similar way to $K(\text{off})$, then the lifetime may remain unchanged. On what basis do you assert that the ADP-bound state of NBD in cyclo is indeed an ADP- P_i -bound state? Do NBD titration with ADP- P_i exhibit the same chemical shift perturbations as in-cyclo? In supplementary figure 5c, residue 145 shows identical CSPs for ADP or ADP- P_i bound states. However, you use this residue as a probe in the in-cyclo experiment with varying P_i concentrations. Why not choose a residue with different chemical shifts in the presence or absence of P_i ? How do V278 or V130 behave during in-cyclo experiments at different P_i concentrations?

p.10-l.285: "The in cyclo experiment thus overcomes the limitations of static experiments."

The "static experiment" meaning is misleading, it would be better to say: "the experiment achieved under either non-equilibrium or equilibrium conditions". Considering that the in-cyclo experiment setting does not allow the ADP to back bind to NBD because it is rapidly converted into ATP, and if we assume that P_i from the bulk cannot interact with NBD without ADP, then one could hypothesize that the presence of P_i in the solution does not affect the release rate of the ADP-bound NBD. Indeed, the Fig. 2b shows that ADP is undetectable in the solution. In that context, you cannot conclude that P_i will not affect the catalytic cycle. The presence of ADP+ P_i in solution can perhaps impact the kinetics of the catalysis, however, in your in-cyclo experimental setting you cannot measure this impact.

p.12-l.325: "These data thus completely exclude any effect of calcium on the native functional cycle, in the cellular concentration range as well as up to several mM."

p.12-l.330: "The data thus clearly establish that calcium ions do bind to the ADP-bound state of the BiP NBD, but not to the ADP- P_i -bound state, and consequently, since the ADP-bound state is not part of the functional cycle of the BiP NBD, physiological variations of calcium have no effect on the BiP NBD functional cycle."

As previously discussed regarding P_i 's role, in the in-cyclo experimental setting, free ADP is absent, preventing ADP back-binding and subsequently, hindering ion binding associated with ADP. In the endoplasmic reticulum (ER), ADP production is not depleted as rapidly as in the in-cyclo setting, allowing for potential effects on ADP back-binding by P_i and Ca concentrations.

Minor concerns:

Oligomeric states have only been assessed for ADP-bound NBD. What about ATP-bound and apo states? Given that BiP oligomerization may involve the NBD surface, one might suspect the homo-association of these domains as well.

The text should undergo a thorough review for consistency in verbal tenses. The authors switch between past and present tenses, which can confuse. Additionally, numerous typos and grammatical errors are present throughout the manuscript, detracting from the reading experience.

Consider providing a more descriptive title that highlights the quantitative characterization and key conclusions of the work. Including a reference to the steady-state setting could also enhance clarity and relevance.

p.3-l.66: "ATP binding leads to a rearrangement of lobe I, which triggers the SBD to open the client binding site. Following ATP hydrolysis, the NBD is in an ADP-bound state that leads to the closure of the client binding site. From there, ADP is released at one point, resulting in the apo form, to which a new ATP molecule binds to restart the cycle."

The catalytic cycle reported here does not inform about the open/close transition that NBD/SBD experiences upon nucleotide binding. In the apo state, NBD and SBD do not interact with each other.

p.3-l.75: "Interestingly, despite the exhaustive biochemical characterization of Hsp70 proteins, measurements of the functional cycle kinetic parameters have so far been possible only by isolating individual steps and not at atomic resolution." How can kinetic parameters be measured at atomic resolution? This sentence is confusing. Are the authors trying to emphasize that the various structural states of the kinetic cycle are not defined at the atomic scale? Please rephrase for clarity.

p.5-l.101: "... the BiP NBD reaches its native state after denaturation and refolding."

This experiment showed similar conformations for both samples (after and before unfolding/refolding). Stating the presence or absence of the native state is an over-interpretation of the results.

p.5-l.103: For coherence, methyl labeling cannot be presented as a new feature here, as it has already been mentioned earlier at p.4-l.98. Please rephrase for consistency.

p.5-l.103: "Ile-[$^{13}\text{C}1\text{H}$] δ 1, Met-[$^{13}\text{C}1\text{H}$] ϵ and Val-[$^{13}\text{C}1\text{H}$] 1/2"

Those chemical names do not follow IUPAC recommendations. By the way, why did you not label Leu methyls, and/or threonine/alanine, also?

p.5-l.111: "... we observed a homogeneous spectrum with a single set of 102 resonances ..."

Did you complete a full titration or confirm that no further evolution of the NMR signal occurred upon adding more ADP/Mg, ensuring protein saturation, and monitoring a pure ADP-Mg-bound state? Also, do you know the affinity constant for this binding event? Is it a fast or slow exchange process compared to the chemical shift time scale?

p.6-l.136: "We selected a buffer composition corresponding to optimal Hsp70 activity ..."

Based on what do you state that those conditions give the highest Hsp70 activity? Please notify.

p.7-l.179: "... under the steady state conditions in cyclo."

This is confusing because one may think there is also a none-steady state in cyclo.

p.8-l.200: "... observed in the lobe IA."

The topology of the protein itself should be displayed (as the different lobes etc.) somewhere in one of the figures (anywhere a structure is displayed).

p.10-l.259: "Strikingly, our measurements show a large and highly significant deviation between the two experiments ($D = 13.6 \pm 2$ s (MABA-ADP) vs $D = 49.7 \pm 3$ s (in cyclo)) (Fig 5a,b). To resolve this conundrum, we realized that the functional state encountered in the functional cycle is the ADP-Pi-bound state, while the MABA-ADP experiment is phosphate-free and its lifetime thus corresponds to the ADP-bound state."

This part is useless, you can delete it and directly show the ADP release rates against the Pi concentration because the impact of phosphate is already known.

Here, I listed some rewording propositions to improve both the clarity and the accuracy of some sentences throughout the manuscript:

p.4-l.78: "... the reaction kinetics of the cycle individual steps." read better as "... the reaction kinetics of each catalytic step."

p.5-l.100: "... the BiP NBD with and without denaturation ..." reads better as "... the BiP NBD before and after denaturation/renaturation ..."

p.5-l.103: "In a next step, we isotope-labelled ..." read better as "In the next step, we isotopically labelled ..."

p.5-l.105: "... allow atomic resolution NMR studies even at large molecular sizes ..." read better as "... allow NMR studies at atomic resolution, even with large macromolecular ..."

p.5-l.108: "With the purification and preparation steps, ..." This sounds useless – I do not understand what the point of stating that.

p.7-l.164: I am not sure about what "classical experiment" stands for. It reads better as "previously published results".

p.7-l.167: "... photospectrometry (A340) to measure the ATP consumption." reads better as "... the absorbance at 340 nm to measure ATP consumption."

p.7-l.168: "... found perfect correspondence within error under the steady state conditions of our in cyclo NMR ..." would read better as "...no significant difference was found with the in cyclo NMR method ..."

p.7-l.170: "... allowing atomic resolution observations." reads better as "... allowing observations at the atomic scale."

There are many typos/grammatical errors all over the manuscript. Here is some proposed corrections, but I stopped correcting them from line 206:

p.1-l.1: "... nucleotide binding ..." read better as "... nucleotide-binding ..."

p.3-l.48: "... and ..." should be "..., and ...".

p.3-l.53: "... that ensure ..." should be "... that ensures ...".

p.3-l.66: "... which takes place ..." read better as "... which take place ...".

p.3-l.70: "... depends on ..." should be "... depends on ...".

p.3-l.73: "... and ..." should be "..., and ...".

p.3-l.75: "... exhaustive ..." should be "... the exhaustive ...".

p.4-l.80: "... in cyclo ..." should be defined before using it.

p.4-l.82: "... independent from ..." should be "... independent of ...".

p.4-l.83: "... characterize absence..." read better as "... is characterized by the absence ..."

p.5-l.109: "... or in presence of ADP." shall be "... or the presence of ADP."

p.5-l.111: "In presence of 5 mM ADP-Pi, ..." shall be "In the presence of 5 mM ADP-Pi, ...".

p.5-l.115: "... single point mutagenesis ..." reads better as "... single-point mutagenesis ...".

p.6-l.123: "... of these network ..." reads better as "... of these networks ..."

p.6-l.129: "... isoleucine, 6/6 methionine ..." reads better as "... isoleucine, and 6/6 methionine ..."

p.7-l.176: "In a next step, ..." reads better as "In the next step, ..."

p.7-l.177: "... get atomic level insights." It reads better as "... get atomic-level insights."

p.7-l.178: "... amounts of the two signals is ..." reads better as "... amounts of the two signals are ..."

p.7-l.183: "... only in presence of the ..." reads better as "... only in the presence of the ..."

p.8-l.201: "... the full-length protein result ..." reads better as "... the full-length protein results ..."

p.8-l.206: "... three commons ATP ..." reads better as "... three common ATP ..."

Reviewer #4

(Remarks to the Author)

Reviewer #5

(Remarks to the Author)

"I co-reviewed this manuscript with one of the reviewers who provided the listed reports. This is part of the Nature Communications initiative to facilitate training in peer review and to provide appropriate recognition for Early Career Researchers who co-review manuscripts."

Version 1:

Reviewer comments:

Reviewer #2

(Remarks to the Author)

The authors have addressed the issues raised previously.

Christian A.M. Wilson and Francesca Burgos-Bravo

Reviewer #3

(Remarks to the Author)

We would like to thank the authors for their detailed responses and substantial improvements to the manuscript, which effectively address my concerns as well as those raised by other reviewers. The extensive effort put into this work has resulted in a comprehensive and well-articulated analysis of the kinetics of BiP's nucleotide-binding domain (NBD). The authors have extended their kinetic framework to incorporate four distinct experimental conditions: first, by introducing a recycling system that removes inorganic phosphate (Pi) as it accumulates in the previous in-cyclo setting, the authors establish a true steady-state regime for NBD activity. Indeed, without adding the second enzyme (E2), which consumes Pi over time, NMR time-course experiments demonstrated that NBD species composition changes progressively as Pi accumulates. Second, by using either E1, E2, or none, the authors record NMR data for four different reaction settings. The enhanced dataset now allows for the determination of all microscopic rate constants (11 in total), providing a complete kinetic characterization of NBD alone and offering a robust foundation for future studies on the influence of additional BiP domains and cofactors on NBD catalytic activity.

Furthermore, the observed distinctions between ATP and ATP analog binding modes are of considerable significance and have broader implications for the general study of ATPases.

The conclusions regarding calcium's impact on BiP kinetics have also been strengthened, offering valuable insights into endoplasmic reticulum chemical conditions that modulate BiP activity. Specifically, the findings highlight that calcium influences BiP kinetics predominantly when ADP concentration is high and Pi is low.

Additionally, the authors have thoroughly clarified the theoretical framework underlying their methodology, facilitating a clear and rapid understanding of the proposed approach and its validation.

With the exception of a few minor concerns, we fully endorse this manuscript for publication in Nature Communications following the suggested revisions.

Figure 6d: The final states, whether apo+Pi or apo+ADP, should have equivalent energies since both configurations correspond to the same chemical state: apo+ADP+Pi. The terminal state of the system remains identical regardless of the reaction pathway - whether ADP release precedes Pi release or vice versa. Hence, the two branches of the reaction scheme ultimately converge to the same thermodynamic endpoint.

Please revise this energy diagram accordingly and explain the dashed-line pathway in the figure caption.

The following are minor suggestions for corrections or comments:

1. Several instances of subject-verb and noun-number agreement require correction.
2. Figure 1c, p.6, l.128, and p.34, l.960: The term "percentage of number of observed NOESY contacts" needs clarification. It seems to refer to the percentage of assigned NOEs, but this definition should be explicitly stated for clarity.
3. Figure 4d: Some residues exhibit sensitivity to ADP·Pi versus ADP, while others do not. Can you confirm whether the residues that detect the additional Pi are localized around its presumed binding site?
4. p.10, l.278: In the single-turnover reaction setup, it is asserted that the product's appearance directly reflects the catalytic rate (k_{cat}). However, this assumption holds true only if the subsequent dissociation steps for ADP and Pi are not significantly slower than the catalytic turnover. If dissociation is rate-limiting, it would dominate the overall reaction rate rather than the catalytic step. Given that the kinetic constants you have determined place dissociation in a similar time range to catalysis, it should not impact your conclusion, but it should be acknowledged. Moreover, the in-cyclo experiment circumvents this issue, a notable advantage worth discussing.
5. Supplementary Figure 14 (upper left): The pictogram appears incorrect, with red colors spreading inconsistently.
6. In your response, you mentioned that functionally deficient mutations of NBD induce substantial conformational rearrangements, detracting from an assignment transfer from apo state to the ATP-bound state. This important point should be explicitly stated in the manuscript, along with references or supporting spectra in the supplementary materials.
7. Consider using "in-cyclo" instead of "in cyclo" throughout the article for improved visual clarity.
8. Lastly, the approach involving numerical integration of non-steady-state experiments alongside in-cyclo experiments raises the question of why both were necessary to derive all kinetic parameters. Would numerical integration of the non-steady-state data alone suffice to determine all 11 rate constants? This fact should be justified in either the introduction or discussion sections.

Reviewer #4

(Remarks to the Author)

Reviewer #5

(Remarks to the Author)

Reviewer #1 (Remarks to the Author):

Mas and Hiller present a new approach to characterize ATPase domains in cyclo with high temporal and spatial resolution. By coupling advanced isotope labeling for methyl-TROSY, atomic-resolution fingerprints of domains in various nucleotide states, and real-time kinetics measurements, the authors gain insight into the functional cycle of an ATPase domain at unprecedented detail. The NMR approach is benchmarked on the Hsp70 nucleotide binding domain (NBD), with the model system being the human ER Hsp70 BiP. The authors compare their in cyclo method to conventional single-turnover ATPase assays and other fluorescence assays for nucleotide binding/dissociation, demonstrating that their in cyclo-NMR method matches the information content from the other assays while also providing atomic-resolution structural insight. Moreover, because the approach outlined by Mas and Hiller relies on an ATP re-generation system, they are able to study the NBD in a more "native-like" nucleotide state -- fluctuating between apo, ATP-bound, ADP-Pi-bound, and repeat. Their data convincingly show that conventional fluorescence-based ADP displacement assays overlook the important effect of the post-hydrolysis state, namely that Pi "locks" ADP into the NBD. This is a central advantage to the method outlined in this work: that the ATPase cycle can be studied without the need for fluorophores or displacement assays. Finally, the authors find no effect for Ca²⁺ on the BiP NBD cycle, suggesting that it remains buffered from Ca²⁺ flux that occurs during ER stress.

Overall, the paper is very clearly written, and the results are logically presented. The data supports the claims in the manuscript, and I recommend the manuscript for publication in Nature Communications following only minor changes.

We thank the reviewer for the appreciation of our work.

My main concern regards the NMR assignments for the ATP-bound state. I've outlined my concerns below, followed by other minor points.

1. Fig 3 – how were the assignments transferred from the ADP-Pi- to the ATP-bound state? The 13 mutants are mentioned, but this set includes only 1 Val residue. "Unambiguous spectral regions" assumes relatively isolated peaks; however, for the crowded Val region this is certainly not trivial (eg Supp Fig 6). Could the authors clarify how the assignments for the ATP-bound state were obtained, especially in the crowded Val region? Were NOEs in the ATP-bound state measured? Was the %T used as an assignment restraint?

Indeed, the transfer of assignments is straightforward for Ile, Met and isolated Val residues and is more complex for Val residues in the spectrally crowded area. We used multiple experiments to confirm the assignment of the ATP-bound state, including single-point mutants and specific labeling. To resolve the specific problem of overlap between valine signals, we used a stereospecific ¹³C¹H- γ ₂-Val-labeled sample. This labelling decreases the number of peaks by two. Using this stereospecific labelling, we could resolve the pairs of peaks belonging together. With these measures, we could unambiguously assign 74 methyl groups in the ATP-bound state, compared to 84 assignments in the ADP-Pi bound state.

The lack of assignment transfer and NMR signal overlap are the main reasons why we have excluded 33 methyl signals from the data analysis in the "in cyclo" experiments. Only the methyl groups for which none of the NMR signals overlaps with other methyl groups signals were included in the population calculation.

We have included these explanations in the revised version of the manuscript.

2. Fig 3D – I agree that the average ADP-Pi % is ~75% over all resonances. But could the authors comment on why there is a spread in the % of the ADP-Pi-bound state of ~65% (V149) to ~90% (V92)? The two states are in slow exchange, so are differential ps-ns or us-ms dynamics within each state responsible for the differences?

In the intensities of the sub-states in cyclo, there is indeed a variation observable among the residues. Regarding the reasons for these variations, we have reached the same conclusions as the reviewer that these are mainly due to differences in local dynamics of individual residues, as well as experimental noise. This variation is the main reason why we base our determination of the population levels not on only a few residues, but on the average of as many independently resolved residues as possible. The resulting average is thus expected to be a highly precise result that cancels out effects of individual local dynamics.

We have added a paragraph in the manuscript to highlight this aspect.

Other points:

Results (Line 122) - 4 NOEs/methyl does not guarantee “unambiguous assignments as a network effect”. This number is a reasonable value for an NOE-based methyl assignment, and fits with about the average number reported by Pritisnac et al. 2017 JACS. However, I don’t think it’s fair to say that 4 NOEs/methyl guarantees an unambiguous assignment, as even these automatic assignment methods based on advanced graph-theory or genetic algorithms cannot guarantee this.

The assignment is based on the combination of single point mutants, NOESY experiments and chemical shift perturbation experiments that match with reported crystal structures. Only those assignments that were without alternative (“unambiguous”) were made. Consequently, there are some missing assignments that we did not make even though ambiguous possibilities would have been available. We thus think it is correct to call our assignment unambiguous, in the tradition of the field.

Results (Line 122) – would the median number of NOEs/methyl be more relevant to report here? How many BiP methyls had fewer than 4 NOEs? How many did not have NOEs? (and if so how were the orphans assigned?)

We have added this information in the material and methods section. As described above, methyl groups assignments that were ambiguous due to a lack of NOEs were not assigned.

Methods (Line 392) – At what temperature was BiP NBD refolded?

The refolding temperature was 4 °C. This has been added to the material and methods section.

Methods (Line 424) – the mutants need clarification. In the early results, lines 115 and 126, it is mentioned that 6 Met were mutated followed by 7 mutations to residues in the core of the methyl-methyl NOE networks. But the Methods (in section line 424) state that one of the mutants failed to express (M148L) and could not be analyzed. So, I suppose then that only 5 of the 6 Met were assigned by mutagenesis and the last one was assigned by process of elimination?

This is correct. We have added this information to the material and methods.

Methods (Line 452) – I assume that the 3D NOESY experiments were fully sampled? (If not, how was the NUS performed and the sparse data reconstructed?)

All 3D NOESY experiments were fully sampled. This is now explicitly stated in the text.

Methods (Line 466) – the addition of co-chaperones and client proteins are mentioned here but I think this line can be removed entirely.

We thank the reviewer for pointing this out. The text has been modified.

Methods (Line 524-5) – is there a reference needed for the ATP/ADP quantification by anion exchange?

We have added a reference.

Fig 1 legend (Line 743) – typo in “right” (righ)

Corrected

Fig 1C – is “%” missing from right-axis label? It says “%” in the legend but not in figure.

Corrected

Fig 1C – how do the authors define the % of number of observed NOESY contacts? (right-axis). I’m not sure what this value means. Eg for 8-10 Å the grey bar shows that ~25% of the NOEs based on the structure are actually observed, but what does the cross at ~90% mean?

The percentage of observed NOESY contacts corresponds to the number of observed NOE contacts (right axis) divided by the number of theoretically expected methyl group within a given distance threshold. For the example of 8–10 Å highlighted by the reviewer, we resolved 90 NOE cross peaks, whereas 380 pairs of methyl groups are within a distance of 8–10 Å in the NBD.ADP.Pi structure, resulting in 24% observed NOESY contacts. As expected, the percentage decreases with increasing intermethyl distance. We have clarified this point in the material and methods.

Fig 3A, add red = ADP-Pi and black = ATP to the spectra for clarity. Especially because Fig 3B has a different spectral coloring scheme that is combined with the red/black text from 3A.

Done

Figure 4A – the legend text is not clear what the data are in reference to (in cyclo or single turnover). It would be helpful to mention that this is a single turnover assay as described in Theyssen et al.

Done

Fig 6A legend does not match the colors shown in the figure. I think 0.001 mM Ca²⁺ is also missing from the legend text.

Corrected

Figure 6B add “no Pi” in purple text to the figure.

Done

Fig 6B legend -- capital "D" for "Data points ..."

Done

Fig 6C legend – delete "and black dash line"

Supp Fig 1c shows the full 2D 1H-13C HMQC of BiP. Why not show this in Figure 1 where assignments are used? Or show the Val region to demonstrate the completeness in assignment.

The assignment of the Val region is provided in Supplementary Figure 3. We chose purposefully to not include the full 2D 1H-13C HMQC of BiP with assignment in the Figure 1 to focus on the more interesting topic of the assignment process instead of the result of the assignment. The complete assignment has been submitted to the BMRB data base.

References to recent (1) real-time NMR work and (2) capturing of an ADP-Pi state on an ATPase could be added: PMID 32790300 and PMID 38326645

We have added these references, and more, in our introduction as example of the usage of NMR to study ATPase.

Line 70 – "dependants" -> depends

Line 79 – individual cycle steps

Line 80 – "ER" already defined

Line 83 – characterize the absence

Line 109 – in the presence of ADP

Line 377 – genscript needs capitalization

Done

Reviewer #2 (Remarks to the Author)

Comments on Mas and Hiller, 2024

The work of Mas and Hiller study the catalytic cycle of ATP in an isolated NBD of human BiP. They use different techniques such as NMR and conventional activity assays (at steady state and single turnover) to understand the step of ATP hydrolysis, ADP-Pi release, etc. They also used different metals, such as calcium, to apply their study in a more physiological state. The main findings are that ADP-Pi release is the rate-limiting step in the functional cycling of BiP NBD, explaining why previous work has not seen the same effect between ATP analogues and ATP, and that neither Calcium nor free inorganic phosphate has a significant effect on its cycling. The paper is very straightforward and well written and I would like to suggest some improvements in the overall story.

Thank you for the appreciation of our work.

I would recommend the authors to include a schematic representation of the Hsp70 cycle, it would be helpful to understand the dynamics and conformational changes of the protein during the functional cycle

The schematic representation of the cycle is now shown in the Figure 2 and Figure 6.

At the end of the introduction, the authors describe the main conclusion of the work, for instance about the effect of calcium. However, the introduction provides very little information about the expected or known effects of calcium. I would recommend to add to include a short background about its reported effects

Thanks for pointing this out - we have updated the text to provide references for the effect of calcium.

In the results section titled 'NMR Resonance Assignment of NBD Methyl Groups', the authors conducted various analyses to confirm the protein's native-like state after refolding. It is unclear from the text whether the protein's activity remained the same before and after the unfolding/refolding cycle.

The ATPase activity of purified and refolded protein is indeed identical within the measurement error. We have now added the comparison in the manuscript (Supplementary Fig. 1b).

Does BiP form a dimer with the NBD domain alone? BiP has been shown to dimerize (Rivera et al., 2023), and it is possible that some dimers could be formed over the NBD itself. Have you checked for the presence of an NBD-NBD dimer in your system?

The oligomeric state of the BiP NBD has been assessed by SEC-MALS experiments at 25 μ M (Supplemental Fig.1 b). The protein is essentially monomeric, dimers were not detected. The absence of oligomers for the isolated NBD is coherent with the literature that describe oligomerization of full-length BiP to be driven by interaction of the BiP SBD with the NBD-SBD disorder linker. Our construct does not contain the SBD and hence this interaction is not expected to be present.

As they mentioned they are using a highly purified BiP, I would recommend adding the % of purity in the SDS-PAGE gel.

We thank the reviewer for the suggestion. The following sentence has been added in the text at the line 98 “The percentage of purity determined by SDS-page gel to be > 98% (Supplementary Fig. 1a)”.

The process of assembling the BiP-ADP-Pi complex is not sufficiently explained.

The NBD-ADP-Pi complex was assembled by incubating NBD with an excess of Pi and highly pure ADP. The Material and Methods section has been modified to include more information regarding the sample preparation.

They verified their NMR assignment by single-point mutagenesis. Did the author confirm that these modifications do not affect the proper folding and activity of the BiP NBD?

Thanks for pointing this out. The proper folding of the mutants was confirmed by their SEC profile and analysis of their NMR fingerprint. Each mutant displayed a large NMR signal dispersion that can be observed only in folded protein. Also, the NMR spectrum of each mutant matched the wildtype spectrum with only few differences, indicating that the effect of the mutation was only local (otherwise, it would anyways have not been possible to use them for assignment).

The ATPase activity of the single-point mutants was not tested, as these mutants were only used for the purpose of NMR resonance assignment and not for structural or functional analysis.

We have added a sentence in the material and methods indicating that the mutants were properly folded.

Is it possible to use NMR data to depict the conformational transition of the BiP NBD domain? For example, in the statement “Additional large chemical shift differences were, however, also observed in lobe IA. These reflect the rotation of lobe IA resulting from ATP binding, which, in the full-length protein, results in docking of the SB” is it only possible to draw this conclusion using other types of structural data, such as cryo or x-ray?

It is in principle possible to obtain precise local structure information by NMR spectroscopy. This can be done by NOE, PRE, PCS and other methods, which however, require intense additional efforts. In the present example, we can explain the observed chemical shift differences with a known conformational transition of the lobe IA and therefore refrain from re-determining this motion. We foresee that in future studies of the full-length BiP there may well be situations where a de novo determination of the local protein structure will be worthwhile.

I recommend labeling the lobes in the 3D structures of the BiP NBD as shown in Figure 3C of the paper.

Thanks for the suggestion. Whenever a BiP NBD 3D structure is presented in the figures, we have included a labeling of the lobes according to the standard nomenclature IA, IIA, IB, IIB.

They state that “The NMR signal intensities of each of the two states is proportional to their relative population and thus to the kinetic rate constants that connect the two states” It would be helpful to include a more detailed explanation of this.

We have added a theory section to explain this in detail, as requested also by referee 3.

In the discussion, they say “Formation of the ADP-Pi post-hydrolysis state has major consequences for the regulation of the functional cycle in response to environmental change”. However, this statement only applies when the NBD domain is alone, which is not the case in the cell. It would be interesting to discuss the possible effects of the SBD and the substrate. Additionally, it is unclear what the authors expect to observe. The study found that Ca does not have any effect on ADP-Pi lifetime. However, it is unclear whether this is due to the absence of the SBD. The discussion section could be improved.

The referee is perfectly right that for full-length BiP, we expect to see biologically interesting new effects from the interplay of NBD and SBD, that are not seen for the NBD. Certainly, the present work will serve as a well-characterized benchmark and starting point for future studies of the more complex system. We have modified the discussion section to highlight this potential. Additionally, we have now added a section to describe in detail how environmental change such as ADP and phosphate concentrations modify the functional cycle.

The discussion section could be improved by correlating the different binding and movement of the NBD domain with specific amino acids in the structure. For instance, Vogel et al. (2006) demonstrated the significance of many amino acids for ATP and ADP-Pi binding, and it would be beneficial to discuss how they vary across different species. Additionally, the authors briefly mention the importance of their findings with the allosteric cycle of BiP, but further elaboration would enhance the discussion. A deeper explanation of the amino acids involved in allosteric communication is recommended to provide more clarity. Predictions could also be made based on this well-known information.

The work by Vogel et al. focuses on the NBD interdomain linker allosteric control and interaction with between the NBD-SBD (PMID 17052976). We have purposefully designed our study on a minimal construct of the NBD that simplifies the functional cycle excluding such complex effects that will be treated in a follow-up study of the full-length BiP functional cycle. We have added a reference in the text to the work of Vogel et al. (2006) and amended the discussion to add more details about the importance of the finding in the context of the allosteric cycle of full-length BiP and its regulation by co-chaperones.

Post-translational modifications of BiP are not mentioned, and now it is an important line of research. I would recommend discussing how different post-translational modification or co-chaperones binding could affect BiP ATP cycle.

We have modified the discussion to highlight potential applications of our experimental setup for studies of PTMs (for example ampylation) and co-chaperone binding.

In lines 214 and 215 the authors mention non-canonical behavior of ATP analogs. This finding is interesting as it has been observed before but not clearly explained, both in vivo and in vitro. It is suggested to discuss this further in the discussion section and correlate it with other papers that show similar behavior (Flynn et al., 1989, Liberek et al., 1991, and from our group Rivera et al., 2023)

We thank the reviewer for pointing this out. We have modified the text accordingly and include the references mentioned.

Minor comments

We thank the reviewer for their careful reading of our manuscript. We have included the following points.

Line 51 In my understanding the abbreviation BiP is Immunoglobulin Binding Protein, that is why the i on BiP is lower case

Line 342 I suggest rephrase “hydrolysis rate determined in cyclo matches the value determined by single turn-over experiments”

Line 350 it say “call” instead of “cell

Line 383 ml should be mL

Line 402 Which method did they use to measure protein concentration?

Line 497-498 and 503 “HKM reaction buffer HKM” seems redundant

Line 502 Instead of “NADH ATP assay” I suggest called, “Coupled ATP assay” or “NADH coupled ATP assay”

Line 505 there is an extra space after the first Sigma-Aldrich

Some of the reagents in Materials and method does not show the origin (as example IPTG)

Reviewer #3 (Remarks to the Author)

The authors present an NMR kinetics analysis of the BiP NBD domain under ATP-regeneration conditions previously described, allowing the study of the enzyme under steady-state conditions. This originality lies in the ability to monitor the visible states of the protein in action at its steady-state regime. They identified two states and assigned them to the ATP-bound and ADP.Pi-bound NBD states. The kinetic parameters align with previous measures, raising questions about the necessity of this method. Finally, they claim that the catalytic kinetics are unaffected by bulk phosphate and calcium, which contradicts prior studies. Moreover, an alternative interpretation of the data can be proposed, making the latter assertions speculative.

We thank the reviewer for their critical assessment of our work and experimental setup. As mentioned by the reviewer, our experimental setup shows excellent agreement with kinetic parameters from classical experiments such as the ATP hydrolysis rate, and these agreements demonstrating the validity of our setup.

The reviewer also highlighted that our experimental setup demonstrates major improvements compared to previous work that justify the necessity of this method, as well as measures kinetic rates that are not accessible otherwise.

Following the reviewer comments, we have extended our in cyclo experimental setup by adding a second enzymatic system that allows to tightly regulate the phosphate concentration in the low μM range. Using this novel setup, we could determine all the kinetic parameters of the functional cycle by kinetic fitting of the experimental data. These parameters allow then to simulate and explore conditions that were previously not accessible using traditional experiments. We further encompass that our experimental setup will be very valuable when used on more complex systems in the future, such as full-length BiP and its co-chaperones.

The method is interesting but lacks clear definitions of kinetic parameters and mathematical formalism, making the conclusions difficult to grasp. Some assertions lack evidence, and alternative explanations could align with the results. The method seems oversold, requiring more rigorous explanations to be persuasive.

Thank you for pointing this out. We have added a theory section to the manuscript.

Additionally, the manuscript is riddled with numerous typos and grammatical errors that need correction.

Nature publishing group does an automated language quality check upon submission and our manuscript was rated 9/10 for language quality, so we think it is not terribly bad. We will of course correct the spotted typos and mistakes from all referees.

Significant revisions are required for the manuscript. Beyond benchmarking the method, the authors do not address common aspects of the ATPase chaperone field, such as the effects of ATP hydrolysis enhancers, ADP exchange factors, or the analysis of various mutants to demonstrate important conformational changes for specific chemical transitions. I suggest that the authors rigorously define the physics and mathematics required for proper data analysis and include more control experiments to minimize ambiguous interpretations.

We very much appreciate the suggestions of the reviewer to include more aspects of the ATPase chaperone field such as JDPs, NEFs, full-length proteins, and mutants. However, as it

becomes apparent, already the content included in the present manuscript requires extended explanation of the length of 6 Figures and 18 Supplementary Figures. Including more experimental work would readily expand the manuscript outside of a reasonable publication size. We will study and report on these more complex systems in the future.

Given that the main claim of this work is to benchmark the steady-state method for use in more complex systems, it might be better suited for a specialized journal focusing on method development.

Due to the potential applications of our experimental setup not only for the Hsp70 field but also for other ATPase, we deem the broad readership of Nature Communication ideal to share our work.

Major Concerns:

1. Precise Definition of Experimental Conditions and Observables:

The manuscript lacks clear definitions of reaction schemes and mathematical frameworks essential for understanding the authors' reasoning and calculations. I recommend providing a reaction scheme, defining both microscopic (associated with single transitions) and macroscopic (apparent) rate constants, and outlining their relationships, possibly in supplementary material. The theoretical enzymology framework, crucial for supporting the authors' hypotheses, is inadequately addressed. Clarification is needed on the distinction between "in cyclo" experiments and steady-state experiments. If they overlap entirely, the rationale for different terminology should be explained.

We have added a theory section that explains all experimental details and mathematical framework to understand the data interpretation. It is correct that the "in cyclo" experiment is a steady-state experiment, but in combination with high-resolution NMR spectroscopy and for molecular machines, we found the terminology "in cyclo NMR" helpful for the discussion of the data and to distinct it from other possible steady-state experiments.

In Figure 7b, the proposed energy diagram, focusing solely on the protein, mistakenly implies equal free energy for the two apo states, overlooking that it pertains to the entire system (enzyme + substrate). This should be corrected for accuracy and could help define various constants and transitions early in the manuscript.

Figure 7b (Figure 6b in the revised manuscript) shows a free energy diagram for non-equilibrium thermodynamics. The different free energies for the apo state occur, because one is apo + ATP and the other is apo + ADP, and this was not explicitly spelled out. The free energy difference is thus arising from the ATP/ADP free energy difference. We expanded the shorthand notation and added text to explain this better.

p.7-l.159 needs a clearer mathematical and theoretical explanation for the calculations of $k(\text{hydr})$ and its relationship with $k(\text{cat}) + k(\text{off ADP})$.

This is included in the new theory section.

p.9-l.226 requires a thorough mathematical analysis to accurately calculate microscopic rate constants, considering the lifetime of ATP-bound NBD and ADP-bound NBD and discussing states like the apo state that may not be visible due to line shape broadening.

The mathematical analysis is now included in the new theory section. The apo state is not visible in the NMR spectra due to the high ATP concentration present in the sample (5 mM), i.e. The ATP on-rate led to fast binding of ATP to the NBD apo. This is now fully explained by the determination of the ATP k_{on}/k_{off} .

2. Critical Evaluation of the Observables:

Questions arise about whether the calculated exchange rates for each transition are consistent with the chemical shift differences between states, indicating slow exchange processes. The absence of visible states in NMR signals, influenced by exchange rates, underscores the need for discussion on dynamic parameters' impact on NMR properties. Different experimental settings could have been employed to vary bulk ATP and ADP concentrations or use mutants to explore the steady-state regime under diverse conditions more thoroughly.

The presence of multiple NMR signals in the in cyclo experiment does not correspond to a slow exchange situation between the same two states in equilibrium. We have added in the new theory section a description how the kinetics are derived from the NMR observables to avoid misunderstanding, and we have added new experiments with variable ATP:ADP ratio in cyclo, batch experiments which go through different ATP:ADP levels over time, and an in cyclo experiment without free phosphate to explore different conditions. Notably, a single set of 11 microscopic kinetic constants fits all experiments simultaneously, and 3 separate independent experiments validate 5 of these constants independently (k_{2off} , k_{3on}/k_{3off} , k_{5off} and k_{cat}).

3. Misleading Statements on Atomic-Scale Structural Information:

CSP analysis provides low-resolution structural insights, influenced by multiple factors such as direct ligand binding or allosteric effects. This complexity challenges precise ligand positioning on a protein surface, rendering CSP analysis a lower-resolution method for this purpose.

We agree with this statement. Please note that we did not use CSPs to position ligands on proteins. We use them to identify the observed states with available X-ray structures, and distinguish different ligand-bound states unambiguously. We have clarified this in text.

4. Demonstrating the Nature of the Unknown State:

Beginning with NMR data on AMPPCP, AMPPNP, and ATP γ S binding before presenting steady-state data would improve clarity. Defining an enzymatic dead mutant could have offered insights into the ATP-bound state, indirectly inferred. The identification of the unknown state, potentially a specific ATP-bound state or an alternative conformation, requires further investigation to clarify its nature during enzyme activity.

We have performed additional experiments in the absence of the ATP regeneration system to show that the signal in question (the ATP-bound state) is populated proportional to the ATP/ADP ratio, thus validating this finding. Furthermore, separate preparations of the apo, Pi-, ADP-, and ADP-Pi-bound states all have different fingerprints. The signal is therefore unambiguously identified as the ATP-bound state. We have included these experiments in the revised manuscript.

Using the non-hydrolysable ATP-analogs to identify the ATP-bound state was something that we indeed had in mind. Our data clearly demonstrate, however, that none of these ATP-analogs

produces the ATP-bound conformation, but rather induces states highly similar to the ADP-bound state. As noted by 2 out of 3 reviewers, this is an important conclusion that is in excellent agreement with the Hsp70 literature (reference 44; 61-64 in the manuscript).

Please note that using a functionally deficient mutant such as T229A or T229G will not help in the identification of the ATP-bound state. Such a mutation in the core of the nucleotide binding site induces large structural re-arrangements. Such structural difference will lead to large chemical shift perturbations that the resulting spectrum will not match the ATP-bound state of the wild type.

Additional issues:

The following are independent major concerns that need to be addressed:

p.9-l.232: "The ATP lifetime in cyclo is identical to classical single turn-over experiments." This chapter presents several issues: Firstly, the statement "Conventional approaches do not allow determination of the ATP hydrolysis rate $k(\text{cat})$ from a steady-state experiment" is incorrect. It is possible to measure $k(\text{cat})$ using a proper setting of the initial conditions of an enzymatic experiment and by defining the reaction scheme, and then, fitting the measured initial velocities against the substrate concentrations.

We agree and have removed this sentence

Secondly, comparing the inverse of the microscopic rate constant $\tau_T = 13.4 \pm 2$ s (in cyclo) with the inverse of the macroscopic rate constant of ATP consumption in a single turnover reaction, $\tau_T = 14.3 \pm 2$ s (single turnover), is invalid. The latter is based on an apparent rate constant influenced by various transitions and experimental conditions, while the former is specific to the hydrolysis step. To compare $k(\text{cat})$ from the NMR steady-state experiment to $k(\text{cat})$ obtained from an established enzyme assay, applying the Michaelis-Menten formalism is initially recommended before considering more complex reaction schemes if significant discrepancies arise between methods.

We show in the new theory section that this comparison is correct.

p.10-l.275: "Because the ADP lifetime in equilibrium experiments is strongly correlated to the bulk phosphate concentration, ..."

Your experiment validates that the $K(\text{off})$ of NBD-bound ADP depends on bulk phosphate concentration, but this doesn't necessarily imply a change in lifetime. If $K(\text{on})$ also depends on Pi concentration in a similar way to $K(\text{off})$, then the lifetime may remain unchanged.

Following this and other comments below, we have substantially expanded this section to resolve the precise mechanism of dissociation of the NBD-ADP- Pi complex. The updated manuscript includes these results and a consistent set of microscopic dissociation constants that explain all experiments simultaneously.

On what basis do you assert that the ADP-bound state of NBD in cyclo is indeed an ADP- Pi -bound state?

The identification of the ADP-Pi bound state during the functional cycle is done by comparison of the chemical shift with independent equilibrium experiments of NBD-ADP, NBD-Pi, and NBD-ADP-Pi. During the revision process, we have implemented a new experimental condition that allows to reveal the exact chemical shift of the NBD-ADP bound state that was missing in the previous version of the manuscript. This shows that the state in cyclo matches exactly the ADP-Pi-bound state and not the ADP-bound state. See Figure 2, 4 and Supplementary Figure 5.

Do NBD titration with ADP-Pi exhibit the same chemical shift perturbations as in-cyclo? In supplementary figure 5c, residue 145 shows identical CSPs for ADP or ADP-Pi bound states. However, you use this residue as a probe in the in-cyclo experiment with varying Pi concentrations. Why not choose a residue with different chemical shifts in the presence or absence of Pi? How do V278 or V130 behave during in-cyclo experiments at different Pi concentrations?

This has been corrected in the new version of the manuscript. We found out that conventional preparation of the ADP-bound sample led to the slow accumulation of free phosphate (by self-hydrolysis within the experimental time). This led us to identify at low Pi concentration the ADP.Pi bound state in the previous manuscript as the ADP bound state.

We have now developed a method that allows to prepare and resolve the pure ADP-bound state in the absence of free phosphate (Figure 2 and Supplementary Figure 4). This is explained in details in the material & methods section. Therefore, we can now resolve the pure ADP-bound state that corresponds to a set of NMR signals that can be separated from the ADP-Pi-bound state during the functional cycle.

p.10-l.285: “The in cyclo experiment thus overcomes the limitations of static experiments.” The “static experiment” meaning is misleading, it would be better to say: “the experiment achieved under either non-equilibrium or equilibrium conditions”.

We thank the reviewer for the suggestions. We have adapted the text accordingly.

Considering that the in-cyclo experiment setting does not allow the ADP to back bind to NBD because it is rapidly converted into ATP, and if we assume that Pi from the bulk cannot interact with NBD without ADP, then one could hypothesize that the presence of Pi in the solution does not affect the release rate of the ADP-bound NBD. Indeed, the Fig. 2b shows that ADP is undetectable in the solution. In that context, you cannot conclude that Pi will not affect the catalytic cycle. The presence of ADP+Pi in solution can perhaps impact the kinetics of the catalysis, however, in your in-cyclo experimental setting you cannot measure this impact.

We thank the referee for pointing us to this new experiment. We have now done experiments with different ADP:ATP ratios, and we have developed an improved in cyclo setup that is completely Pi-free. All experimental datasets (Supplementary Fig. 10) can be explained by a single set of microscopic constants and they also resolve the mechanism of dissociation of the NBD-ADP-Pi complex. This aspect was indeed shortened out in the previous version of the manuscript but is now well-resolved.

We have then used the kinetic parameters resolved by the fitting of our experimental data to simulate different conditions that mimic either the cytoplasm or endoplasmic reticulum offering a complete description of the functional cycle under different conditions (Supplementary Figure 14).

p.12-l.325: “These data thus completely exclude any effect of calcium on the native functional cycle, in the cellular concentration range as well as up to several mM.”

p.12-l.330: “The data thus clearly establish that calcium ions do bind to the ADP-bound state of the BiP NBD, but not to the ADP-Pi-bound state, and consequently, since the ADP-bound state is not part of the functional cycle of the BiP NBD, physiological variations of calcium have no effect on the BiP NBD functional cycle.”

As previously discussed regarding Pi's role, in the in-cyclo experimental setting, free ADP is absent, preventing ADP back-binding and subsequently, hindering ion binding associated with ADP. In the endoplasmic reticulum (ER), ADP production is not depleted as rapidly as in the in-cyclo setting, allowing for potential effects on ADP back-binding by Pi and Ca concentrations.

We have done new experiments with different ADP concentrations, and new in cyclo experiments that are completely Pi-free. We present these results in the Fig. 5 and Supplementary Fig. 17,18, allowing to show that indeed the ADP back-binding leads to the formation of the ADP.Ca bound state and the formation of this state is ADP and phosphate dependent. Our data therefore provide a novel and quantitative model of how calcium concentration modifies the length of the cycle by re-binding of ADP under high ADP and low Pi concentrations.

Minor concerns:

Oligomeric states have only been assessed for ADP-bound NBD. What about ATP-bound and apo states? Given that BiP oligomerization may involve the NBD surface, one might suspect the homo-association of these domains as well.

The oligomeric state of the BiP NBD was assessed by SEC-MALS experiments (Supplemental Fig.1 c). The protein is monomeric, dimers were not detected. The literature describes BiP oligomerisation to be driven by interaction of the BiP SBD with the NBD-SBD disorder linker. Our construct does not contain the SBD and hence oligomerization is not expected to be present.

The text should undergo a thorough review for consistency in verbal tenses. The authors switch between past and present tenses, which can confuse. Additionally, numerous typos and grammatical errors are present throughout the manuscript, detracting from the reading experience.

We have done our best to address these points.

Consider providing a more descriptive title that highlights the quantitative characterization and key conclusions of the work. Including a reference to the steady-state setting could also enhance clarity and relevance.

p.3-l.66: “ATP binding leads to a rearrangement of lobe I, which triggers the SBD to open the client binding site. Following ATP hydrolysis, the NBD is in an ADP-bound state that leads to the closure of the client binding site. From there, ADP is released at one point, resulting in the apo form, to which a new ATP molecule binds to restart the cycle.”

The catalytic cycle reported here does not inform about the open/close transition that NBD/SBD experiences upon nucleotide binding. In the apo state, NBD and SBD do not interact with each other.

Thanks for these suggestions, we have modified the title of the manuscript. This part of the introduction provides to the non-specialist reader an overview on the current literature of

Hsp70 that is needed to understand the results presented in the manuscript and potential application of the in cyclo experiment. This is necessary even if we do not report here in the full-length BiP. We agree with the reviewer that in the apo state, or ADP state, the NBD and SBD do not interact with each other.

p.3-l.75: “Interestingly, despite the exhaustive biochemical characterization of Hsp70 proteins, measurements of the functional cycle kinetic parameters have so far been possible only by isolating individual steps and not at atomic resolution.”

How can kinetic parameters be measured at atomic resolution? This sentence is confusing. Are the authors trying to emphasize that the various structural states of the kinetic cycle are not defined at the atomic scale? Please rephrase for clarity.

We have changed the text to avoid the confusion reported by the reviewer.

p.5-l.101: “... the BiP NBD reaches its native state after denaturation and refolding.”

This experiment showed similar conformations for both samples (after and before unfolding/refolding). Stating the presence or absence of the native state is an over-interpretation of the results.

As the refolded protein is indistinguishable from the native state, by the atomic-resolution method NMR, it is fair to say it has reached the same state.

p.5-l.103: For coherence, methyl labeling cannot be presented as a new feature here, as it has already been mentioned earlier at p.4-l.98. Please rephrase for consistency.

Thanks for pointing this out. The text has been changed accordingly.

p.5-l.103: “Ile-[13C1H] δ 1, Met-[13C1H] ϵ and Val-[13C1H] γ 1/ γ 2”

Those chemical names do not follow IUPAC recommendations. By the way, why did you not label Leu methyls, and/or threonine/alanine, also?

These are common names used in the NMR field to describe these methyl groups (Gans et al. 2010, Kerfah et al. 2015, Tugarinov et al. 2015).

It would be not per se a problem to extend the study to threonine/alanine/leucine methyls, but we felt the number of probes reached with the current labelling is sufficiently large to overdetermine the conclusions.

p.5-l.111: “... we observed a homogeneous spectrum with a single set of 102 resonances ...”

Did you complete a full titration or confirm that no further evolution of the NMR signal occurred upon adding more ADP/Mg, ensuring protein saturation, and monitoring a pure ADP-Mg-bound state? Also, do you know the affinity constant for this binding event? Is it a fast or slow exchange process compared to the chemical shift time scale?

In the equilibrium experiments, ADP was present either at 1 or 5 mM concentration, i.e. in a 10 to 50-fold excess relatively to the NBD. We have measured the ADP affinity for NBD by ITC experiments. The K_D is determined to around 1 μ M (Supplementary Figure 4d and Supplementary Table 1). Accordingly, in the equilibrium saturation of the protein is fully achieved in presence of 1 or 5 mM ADP.

p.6-l.136: “We selected a buffer composition corresponding to optimal Hsp70 activity ...”
Based on what do you state that those conditions give the highest Hsp70 activity? Please notify.

We did not write highest Hsp70 activity but “optimal Hsp70 activity”. The HKM (Hepes/KCl/Mg) buffer is a standard buffer used for the study of Hsp70 since the early 1990ies. The text has been modified with a reference to the literature these conditions are based on (45,54-56)

p.7-l.179: “... under the steady state conditions in cyclo.”
This is confusing because one may think there is also a none-steady state in cyclo.

Thanks, the text has been modified.

p.8-l.200: “... observed in the lobe IA.”
The topology of the protein itself should be displayed (as the different lobes etc.) somewhere in one of the figures (anywhere a structure is displayed).

All Figures have been updated to include the NBD topology.

p.10-l.259: “Strikingly, our measurements show a large and highly significant deviation between the two experiments ($\tau_D = 13.6 \pm 2$ s (MABA-ADP) vs $\tau_D = 49.7 \pm 3$ s (in cyclo)) (Fig 5a,b). To resolve this conundrum, we realized that the functional state encountered in the functional cycle is the ADP-Pi-bound state, while the MABA-ADP experiment is phosphate-free and its lifetime thus corresponds to the ADP-bound state.”
This part is useless, you can delete it and directly show the ADP release rates against the Pi concentration because the impact of phosphate is already known.

We have modified the text and figures accordingly.

Here, I listed some rewording propositions to improve both the clarity and the accuracy of some sentences throughout the manuscript:

Thanks, we have adopted these changes.

p.4-l.78: “... the reaction kinetics of the cycle individual steps.” read better as “... the reaction kinetics of each catalytic step.”

p.5-l.100: “... the BiP NBD with and without denaturation ...” reads better as “... the BiP NBD before and after denaturation/renaturation ...”

p.5-l.103: “In a next step, we isotope-labelled ...” read better as “In the next step, we isotopically labelled ...”

p.5-l.105: “... allow atomic resolution NMR studies even at large molecular sizes ...” read better as “... allow NMR studies at atomic resolution, even with large macromolecular ...”

p.5-l.108: “With the purification and preparation steps, ...” This sounds useless – I do not understand what the point of stating that.

p.7-l.164: I am not sure about what “classical experiment” stands for. It reads better as “previously published results”.

p.7-l.167: “... photospectrometry (A340) to measure the ATP consumption.” reads better as “... the absorbance at 340 nm to measure ATP consumption.”

p.7-l.168: “... found perfect correspondence within error under the steady state conditions of our in cyclo NMR ...” would read better as “...no significant difference was found with the in cyclo NMR method ...”

p.7-l.170: “... allowing atomic resolution observations.” reads better as “... allowing

observations at the atomic scale.”

There are many typos/grammatical errors all over the manuscript. Here is some proposed corrections, but I stopped correcting them from line 206:

- p.1-l.1: “... nucleotide binding ...” read better as “... nucleotide-binding ...”
- p.3-l.48: “... and ...” should be “..., and ...”.
- p.3-l.53: “... that ensure ...” should be “... that ensures ...”.
- p.3-l.66: “... which takes place ...” read better as “... which take place ...”.
- p.3-l.70: “... dependents on ...” should be “... depends on ...”.
- p.3-l.73: “... and ...” should be “..., and ...”.
- p.3-l.75: “... exhaustive ...” should be “... the exhaustive ...”.
- p.4-l.80: “... in cyclo ...” should be defined before using it.
- p.4-l.82: “... independent from ...” should be “... independent of ...”.
- p.4-l.83: “... characterize absence...” read better as “... is characterized by the absence ...”
- p.5-l.109: “... or in presence of ADP.” shall be “... or the presence of ADP.”.
- p.5-l.111: “In presence of 5 mM ADP-Pi, ...” shall be “In the presence of 5 mM ADP-Pi, ...”.
- p.5-l.115: “... single point mutagenesis ...” reads better as “... single-point mutagenesis ...”.
- p.6-l.123: “... of these network ...” reads better as “... of these networks ...”
- p.6-l.129: “... isoleucine, 6/6 methionine ...” reads better as “... isoleucine, and 6/6 methionine ...”
- p.7-l.176: “In a next step, ...” reads better as “In the next step, ...”
- p.7-l.177: “... get atomic level insights.” It reads better as “... get atomic-level insights.”
- p.7-l.178: “... amounts of the two signals is ...” reads better as “... amounts of the two signals are ...”
- p.7-l.183: “... only in presence of the ...” reads better as “... only in the presence of the ...”
- p.8-l.201: “... the full-length protein result ...” reads better as “... the full-length protein results ...”
- p.8-l.206: “... three commons ATP ...” reads better as “... three common ATP ...”

REVIEWERS' COMMENTS

Reviewer #2 (Remarks to the Author):

The authors have addressed the issues raised previously.

Christian A.M. Wilson and Francesca Burgos-Bravo

We thank the reviewers for the appreciation of our work.

Reviewer #3 (Remarks to the Author):

We would like to thank the authors for their detailed responses and substantial improvements to the manuscript, which effectively address my concerns as well as those raised by other reviewers. The extensive effort put into this work has resulted in a comprehensive and well-articulated analysis of the kinetics of BiP's nucleotide-binding domain (NBD).

The authors have extended their kinetic framework to incorporate four distinct experimental conditions: first, by introducing a recycling system that removes inorganic phosphate (Pi) as it accumulates in the previous in-cyclo setting, the authors establish a true steady-state regime for NBD activity. Indeed, without adding the second enzyme (E2), which consumes Pi over time, NMR time-course experiments demonstrated that NBD species composition changes progressively as Pi accumulates. Second, by using either E1, E2, or none, the authors record NMR data for four different reaction settings. The enhanced dataset now allows for the determination of all microscopic rate constants (11 in total), providing a complete kinetic characterization of NBD alone and offering a robust foundation for future studies on the influence of additional BiP domains and cofactors on NBD catalytic activity.

Furthermore, the observed distinctions between ATP and ATP analog binding modes are of considerable significance and have broader implications for the general study of ATPases.

The conclusions regarding calcium's impact on BiP kinetics have also been strengthened, offering valuable insights into endoplasmic reticulum chemical conditions that modulate BiP activity. Specifically, the findings highlight that calcium influences BiP kinetics predominantly when ADP concentration is high and Pi is low. Additionally, the authors have thoroughly clarified the theoretical framework underlying their methodology, facilitating a clear and rapid understanding of the proposed approach and its validation.

With the exception of a few minor concerns, we fully endorse this manuscript for publication in Nature Communications following the suggested revisions.

We thank the reviewer for the appreciation of our work and their suggestions.

Figure 6d: The final states, whether apo+Pi or apo+ADP, should have equivalent energies since both configurations correspond to the same chemical state: apo+ADP+Pi. The terminal state of the system remains identical regardless of the reaction pathway - whether ADP release precedes Pi release or vice versa. Hence,

the two branches of the reaction scheme ultimately converge to the same thermodynamic endpoint.

Please revise this energy diagram accordingly and explain the dashed-line pathway in the figure caption.

Thanks for pointing this out. We had previously ended the two parallel pathways in two different states, apo+ADP and apo+Pi, with different free energies (the respective other ligand is removed by the enzymatic system), so the Figure was correct. Nonetheless, we like to follow the reviewer's suggestion and now end both pathways in the apo+ADP+Pi state.

The following are minor suggestions for corrections or comments:
1. Several instances of subject-verb and noun-number agreement require correction.

We thank the reviewer for their careful reading of our manuscript. We have addressed this comment by additional proof-reading of the text.

2. Figure 1c, p.6, l.128, and p.34, l.960: The term "percentage of number of observed NOESY contacts" needs clarification. It seems to refer to the percentage of assigned NOEs, but this definition should be explicitly stated for clarity.

We thank the reviewer for pointing this out. We have revised this Figure to improve clarity.

3. Figure 4d: Some residues exhibit sensitivity to ADP·Pi versus ADP, while others do not. Can you confirm whether the residues that detect the additional Pi are localized around its presumed binding site?

The adaptation of the NBD to the additional Pi results in signal perturbations both in the proximity of the binding site as well as elsewhere, due to allosteric structural changes. We show them collectively in Figure 4d, but did not further interpret them in the present work.

4. p.10, l.278: In the single-turnover reaction setup, it is asserted that the product's appearance directly reflects the catalytic rate (k_{cat}). However, this assumption holds true only if the subsequent dissociation steps for ADP and Pi are not significantly slower than the catalytic turnover. If dissociation is rate-limiting, it would dominate the overall reaction rate rather than the catalytic step. Given that the kinetic constants you have determined place dissociation in a similar time range to catalysis, it should not impact your conclusion, but it should be acknowledged. Moreover, the in-cyclo experiment circumvents this issue, a notable advantage worth discussing.

The single-turnover experiment starts from a batch of uniform, stoichiometrically formed ATP.NBD complex. The kinetics of catalysis can indeed be perturbed by ligand dissociation, but mostly by product dissociation rather than by educt dissociation. The effective rate that is measured at low protein concentrations is $k_{cat}^* = k_{cat} + k_{off1}$ and under our conditions $k_{cat}^* \approx k_{cat}$. We have added an explanation to the main text and to the theory section to highlight this point and adapted the discussion.

5. Supplementary Figure 14 (upper left): The pictogram appears incorrect, with red colors spreading inconsistently.

We thank the reviewer for noticing this and have corrected this inconsistency.

6. In your response, you mentioned that functionally deficient mutations of NBD induce substantial conformational rearrangements, detracting from an assignment transfer from apo state to the ATP-bound state. This important point should be explicitly stated in the manuscript, along with references or supporting spectra in the supplementary materials.

Our comment referred to full-length BiP, for which we have extensively characterized the functionally deficient mutant T229G/A. We will present these data in an upcoming publication concerning full-length BiP. The corresponding NBD mutant T229G/A has been challenging to purify, hence we have no data to expand on this topic in the present manuscript.

7. Consider using “in-cyclo” instead of “in cyclo” throughout the article for improved visual clarity.

Thanks for the recommendation, which we have implemented.

8. Lastly, the approach involving numerical integration of non-steady-state experiments alongside in-cyclo experiments raises the question of why both were necessary to derive all kinetic parameters. Would numerical integration of the non-steady-state data alone suffice to determine all 11 rate constants? This fact should be justified in either the introduction or discussion sections.

Thanks for pointing this out. Fitting of the non-steady-state data alone did not lead to a stable solution for the set of 11 rate constants. We mention this now in the discussion section.

Reviewer #4 (Remarks to the Author):

Reviewer #5 (Remarks to the Author):
